# The scaffolding protein flot2 promotes cytoneme-based transport of wnt3 in gastric cancer

**Daniel Routledge[1], Sally Rogers[1], Yosuke Ono[1], Lucy Brunt[1], Valerie Meniel[2], Giusy Tornillo[2], Hassan Ashktorab[3], Toby J Phesse[2,4], Steffen Scholpp[1]\***

[1]Living Systems Institute, School of Biosciences, College of Life and Environmental Sciences, University of Exeter, Exeter, United Kingdom; [2]The European Cancer Stem Cell Research Institute, School of Biosciences, Cardiff University, Cardiff, United Kingdom; [3]Department of Medicine, Howard University, Washington, United States; [4]The Peter Doherty Institute for Infection and Immunity, The University of Melbourne, Melbourne, Australia

**Abstract** The Wnt/β-catenin signalling pathway regulates multiple cellular processes during development and many diseases, including cell proliferation, migration, and differentiation. Despite their hydrophobic nature, Wnt proteins exert their function over long distances to induce paracrine signalling. Recent studies have identified several factors involved in Wnt secretion; however, our understanding of how Wnt ligands are transported between cells to interact with their cognate receptors is still debated. Here, we demonstrate that gastric cancer cells utilise cytonemes to transport Wnt3 intercellularly to promote proliferation and cell survival. Furthermore, we identify the membrane-bound scaffolding protein Flotillin-2 (Flot2), frequently overexpressed in gastric cancer, as a modulator of these cytonemes. Together with the Wnt co-receptor and cytoneme initiator Ror2, Flot2 determines the number and length of Wnt3 cytonemes in gastric cancer. Finally, we show that Flotillins are also necessary for Wnt8a cytonemes during zebrafish embryogenesis, suggesting a conserved mechanism for Flotillin-mediated Wnt transport on cytonemes in development and disease.

**\*For correspondence:**
s.scholpp@exeter.ac.uk

**Competing interest:** The authors declare that no competing interests exist.

## Editor's evaluation

This work should be of interest to cell and developmental biologists in the Wnt and cytoneme fields. The authors convincingly demonstrate that the membrane tethered scaffolding protein Flotillin-2 localizes to and stimulates Wnt cytoneme growth in gastric cancer cells and Zebrafish. This study extends previous cytoneme studies and provides new details about conserved regulatory events controlling Wnt ligand distribution.

## Introduction

Wnt/β-catenin signalling activity has been well-characterised in the glands of the gastrointestinal epithelium (*Clevers, 2013*). In these intestinal crypts, the expression of Wnt/β-catenin target genes occurs in a gradient, with the highest activity at the base of the crypt (*Muñoz et al., 2012*). Here, Paneth cells are located between the intestinal stem cells and regulate proliferation by producing high levels of Wnt3 (*Farin et al., 2012*; *Sato et al., 2011*). Genomic analysis suggests that in about 50% of gastric tumours, the Wnt/β-catenin pathway is deregulated (*Koushyar et al., 2020*).

In many gastric carcinomas, Wnt3 expression is elevated compared with normal gastric epithelium, leading to increased proliferation in the tumour mass (*Voloshanenko et al., 2013*; *Wang et al., 2016*). In development, Wnt proteins can spread over a distance of many cells despite their hydrophobic nature (*Routledge and Scholpp, 2019*). However, how lipid-modified Wnt3 is disseminated in gastric tumour tissues is unclear.

Recently, thin and actin-rich membranous protrusions known as cytonemes have been identified as crucial for transporting essential signalling components between cells (*Kornberg and Roy, 2014*; *Zhang and Scholpp, 2019*). For example, in zebrafish embryogenesis, cytonemes are highly dynamic. They have been reported to transport Wnt components (*Stanganello et al., 2015*), while in the chick limb bud, cytonemes transport Shh (*Sanders et al., 2013*). Moreover, Wnt/PCP signalling in the source cell facilitates the emergence of long and stable Wnt-transporting cytonemes, promoting paracrine Wnt signal activation (*Mattes et al., 2018*; *Brunt et al., 2021*).

Similar to Wnt3, Flot2 is linked to tumour development and proliferation, and its expression is increased in gastric cancer (GC) (*Cao et al., 2013*; *Zhu et al., 2013*). Flotillins are highly conserved, membrane-bound scaffolding proteins, which cluster in detergent-resistant, cholesterol-rich regions of the membrane (*Lang et al., 1998*; *Stuermer et al., 2001*). These Flotillin microdomains act as platforms to facilitate protein-protein interactions and regulate growth factor signalling (*Banning, 2014*). In *Drosophila*, Flotillin-2/Reggie-1 (Flot2) promotes long-range spreading and signalling of the lipophilic ligands, Wingless (Wg) and Hedgehog (Hh) (*Katanaev et al., 2008*), and Flot2 can promote the formation of Hh cytonemes (*Bischoff et al., 2013*). In cancer cell lines, Flot2 also facilitates the formation of filopodia-like protrusions via direct regulation of the actin cytoskeleton (*Neumann-Giesen et al., 2004*; *Gauthier-Rouvière et al., 2020*).

In this study, we found that human GC cells use cytonemes to disseminate Wnt3 proteins intercellularly. We further detected that these Wnt3-bearing cytonemes are dependent on Flot2 function. Our data suggest that the Flot2 function is required for intracellular trafficking and signalling of the Wnt co-receptor Ror2, a key regulator of cytoneme formation. In accordance with our results in GC cells, we identified Flotillins as modulators of Wnt8a cytoneme emergence in zebrafish development, suggesting a conserved role for Flot2 in Wnt cytoneme emergence in vertebrates.

## Results

### Gastric cancer cells utilise cytonemes for paracrine Wnt3 signalling

First, we examined the gastric adenocarcinoma cell lines MKN-28, MKN-7, and AGS and compared these to a gastric epithelial cell line HFE-145 for their potential to form actin-based protrusions. All GC cells form many dynamic actin-based filopodia, and AGS cells display the longest filopodia (*Figure 1A and B*). Next, we asked if GC cells use these filopodial protrusions to mobilise Wnt3 ligands. Using our advanced filopodia fixation and immunohistochemistry (IHC) protocol (*Rogers and Scholpp, 2021*), we observed that HFE-145 have numerous protrusions, which are decorated with Wnt3 (*Figure 1C*). In the cancer cell lines, Wnt3 localisation was enriched on these filopodia and more often detectable (*Figure 1C and D*). Specifically, the AGS line shows an increased number of Wnt3 cytonemes (*Figure 1D*); therefore, we used AGS in the further analysis. Over-expression of Wnt3 in AGS led to an even further increased Wnt3 staining on more filopodia (*Figure 1D and E*). With the same fixation protocol and imaging settings, we could not detect Wnt3 localisation at the plasma membrane and on filopodia after treatment of AGS cells with the porcupine (PORCN) inhibitor IWP2 (*Figure 1F*), suggesting that PORCN-dependent palmitoylation is essential for localisation of Wnt3 on these actin-based protrusions. Similar to the endogenous protein, we found that Wnt3-mCherry is transported along filopodia and localises to their tips (*Figure 1G*, *Figure 1—video 1*).

In addition to proteins that are associated with the actin core, filopodia also contain actin-based molecular motors. In particular, Myosin-X (MyoX) is localised to the tips of the filopodia and is known to facilitate filopodia growth (*Berg and Cheney, 2002*; *Bohil et al., 2006*). Furthermore, MyoX has been shown to be involved in Shh cytoneme function (*Hall et al., 2021*). IHC analysis revealed that AGS cytonemes are decorated with MyoX (*Figure 1H*). After covalent attachment of a mono-unsaturated palmitoleate, Wnts can bind to Evi/Wntless in the ER to facilitate intracellular transport to the plasma membrane (*Bartscherer et al., 2006*; *Bänziger et al., 2006*). Therefore, we asked if Evi/Wntless is also localised on Wnt3 signalling filopodia. First, we observed that Evi/Wntless could be found on

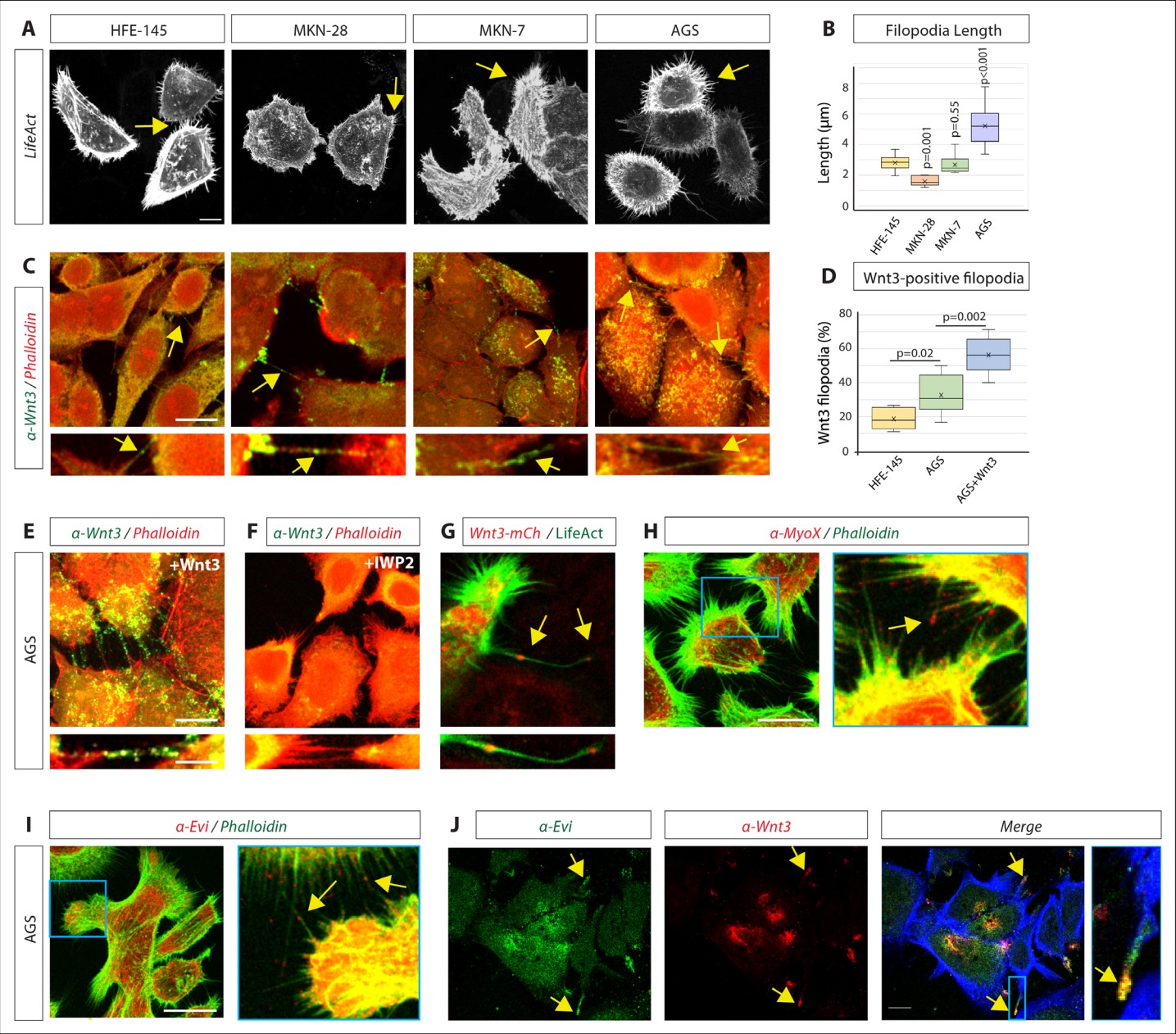

**Figure 1.** Gastric epithelial normal and cancer cell lines utilise cytonemes to transport Wnt3 intercellularly. (**A**) Confocal images of normal gastric epithelial cell line (HFE-145) and gastric cancer (GC) cell lines (MKN-28, MKN-7, and AGS) expressing LifeAct-GFP to visualise actin-based structures. Yellow arrows indicate examples of filopodia. (**B**) Quantification of filopodia length in GC cell lines MKN-28, MKN-7, and AGS (n=7, 8, 25; n=number of cells). Significance is calculated by Student's t-test. (**C**) Immunofluorescent images of HFE, MKN-28, MKN-7, and AGS, stained with antibodies against Wnt3 (green) and actin (Phalloidin-iFluor594, red). Scale bar 10 µm. High-magnification images indicate an example of a Wnt3-bearing cytonemes. Scale bar 2.5 µm. (**D**) Quantification of Wnt3-positive filopodia in gastric epithelial (HFE-145) and cancer (AGS) cells as a percentage of total filopodia (number of cells analysed = 6, 8, 6). Significance is calculated by Student's t-test. (**E**) Immunohistochemistry (IHC) images of AGS cells overexpressing Wnt3 and stained with an antibody against Wnt3 (green) and actin (iFluor594, red). Scale bar 10µm. High-magnification images highlight cytonemes. Scale bar 2.5 µm. (**F**) IHC images of AGS cells treated with the Porcupine inhibitor IWP2 (100 µM, 48 hr) and stained with an antibody against Wnt3 (green) and actin (iFluor594, red). (**G**) Live confocal cell imaging of AGS cells expressing Wnt3-mCherry and LifeAct-GFP. Cytoneme-localised Wnt3-mCherry highlighted by yellow arrows. (**H–J**) IHC images of AGS cells stained with antibodies against (**H**) Myosin-X (MyoX) and (**I**) Evi/Wntless (red) and Wnt3 (red) and (**J**) Evi/Wntless (green Scale bars 10 µm). Phalloidin labels actin (FITC-Phalloidin, green; Phalloidin-iFluor350, blue).

The online version of this article includes the following video and source data for figure 1:

**Source data 1.** Western blot images of HFE, AGS, MKN7, and MKN28 cell lysates stained for Flot2.

**Figure 1—video 1.** Wnt3 transport on cytonemes.

https://elifesciences.org/articles/77376/figures#fig1video1

many filopodia of AGS cells (*Figure 1I*). Furthermore, we found that Wnt3 and Evi/Wntless co-localise on these filopodia (*Figure 1J*). We conclude that Wnt3 can be loaded on the filopodia-like protrusion, which we termed cytonemes, based on their definition as signalling filopodia (*Kornberg and Roy, 2014*; *Mattes et al., 2018*; *Zhang and Scholpp, 2019*).

## Wnt3 cytonemes regulate paracrine Wnt/β-catenin signalling in gastric cancer cells

Next, we assessed the functional impact of Wnt3 cytonemes on paracrine Wnt signalling in GC cells by performing a cocultivation assay of Wnt-producing AGS cells and Wnt-receiving HFE-145 cells (*Figure 2A*). Here, HFE-145 cells were transfected with the Wnt SuperTOPFlash (STF) reporter, while AGS cells were transfected with GFP-tagged constructs to manipulate Wnt3 cytonemes. Indeed, increased Wnt3 can be detected in HFE-145 expressing the reporter in co-culture with AGS cells compared to HFE-145 alone (*Figure 2—figure supplement 1A*). To specifically block cytoneme formation, we used a dominant-negative mutant of the insulin receptor tyrosine kinase substrate p53 (IRSp53). IRSp53 is a multidomain BAR protein which binds active Cdc42 and N-Wasp to promote filopodia formation (*Kast et al., 2014*). Here, the mutated protein IRSp53$^{4K}$ blocks filopodia formation (*Figure 2—figure supplement 1B, C Meyen et al., 2015*) without interfering with autocrine Wnt signalling (*Stanganello et al., 2015*; *Brunt et al., 2021*). Expression of IRSp53$^{4K}$-GFP in AGS cells significantly reduced paracrine Wnt STF reporter activation (*Figure 2B and C*). Transfection of Wnt3 in the source cells doubled STF reporter activity in the cocultivated HFE-145 cells, which could be attenuated significantly by the cotransfection of IRSp53$^{4K}$-GFP. Our data suggest that cytonemes are vital for paracrine Wnt/β-catenin signal activation. Therefore, we conclude that Wnt3 protein can be disseminated via IRSp53-dependent cytonemes in GC cells.

The Wnt/β-catenin signalling pathway is essential for the proliferation and self-renewal of both cancer stem and progenitor cells (*Koushyar et al., 2020*). To assess the impact of paracrine Wnt signalling on cell proliferation, the number of HFE-145 cells after co-cultivation with AGS cells was counted (*Figure 2D*). Here, we found that transfection of Wnt3 in AGS cells caused a substantial increase in HFE-145 cell number. Consistently, blockage of cytoneme formation in AGS cells and Wnt3 overexpressing AGS cells by IRSp53$^{4K}$ cotransfection reduced HFE-145 cell number significantly. Next, we confirmed that the observed alteration in cell numbers is based on proliferation by performing a BrdU incorporation assay to quantify actively proliferating cells with newly synthesised DNA (*Figure 2E*). We over-expressed Wnt3 in AGS cells and co-cultivated them with HFE-145 cells (as in *Figure 2A*). Here, we found a significant increase in the percentage of BrdU-positive cells (*Figure 2E and F*). Consistently, we observed that blockage of cytoneme formation by over-expressing IRSp53$^{4K}$ caused a reduction in proliferating cells. In Wnt3 over-expressing conditions, IRSp53$^{4K}$ showed a modest attenuation of proliferation (*Figure 2F*), suggesting that HFE-145 cells, which are not expressing IRSp53$^{4K}$, can still form cytonemes, and may also be contributing to the proliferative rates.

The single-cell colony formation (CF) assay tests the ability of cancer stem cells to generate a new colony from a single cell. We performed this CF assay to determine if paracrine Wnt3 was being transmitted by cytonemes to functionally regulate the ability of receiving AGS cells to activate cancer stem cell activity. Therefore, we transfected AGS cells with Wnt3 and IRSp53$^{4K}$, mixed these cells with AGS cells stably transfected with red fluorescent protein (RFP) and maintained the co-cultivation for two days to allow Wnt3 signalling between the co-cultured cells. Fluorescence-activated cell sorting (FACS) was used to purify and isolate individual red-fluorescent AGS cells (Wnt3-receivers) for CF over the following 8–10 days. We observed that Wnt3-transfected AGS cells induced more colonies in the receiving cells than control AGS cells (*Figure 2G and H*). In addition, we found that inhibition of cytoneme formation via co-expression of IRSp53$^{4K}$ in Wnt3-positive AGS cells leads to a significant reduction of induced colonies compared to Wnt3 expression only. Our data provide the first evidence that Wnt3 localises to cytonemes and cytoneme-delivered Wnt3 can induce paracrine Wnt/β-catenin transduction cascade to promote cancer stem cell activity and proliferation of GC cells.

## Flot2 expression level correlates with cytoneme phenotypes in gastric cancer cells

To compare the molecular mechanism concerning the cytoneme-mediated Wnt3 dissemination in gastric epithelium cells and GC cells, we investigated the function of Flot2, a scaffolding protein,

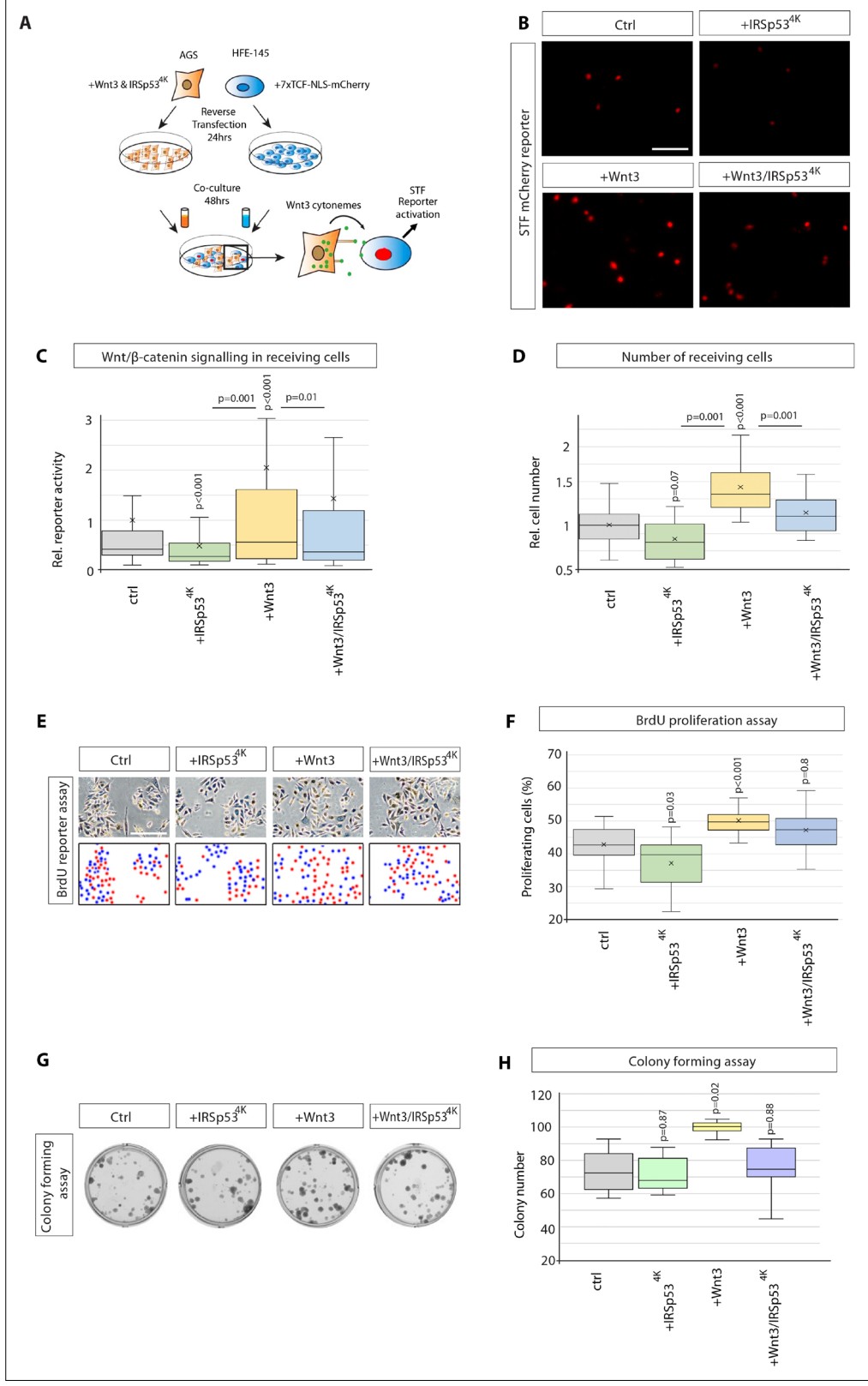

**Figure 2.** Wnt3 cytonemes regulate paracrine Wnt/β-catenin signalling and proliferation. (**A**) Experimental protocol for measuring paracrine Wnt signalling activation. HFE cells expressing the SuperTOPFlash (STF) reporter, 7×TCF-NLS-mCherry, were cocultivated with AGS cells expressing indicated constructs. Fluorescence of STF mCherry reporter was measured after 48 hr and compared to untransfected control cells. (**B**) Representative

*Figure 2 continued on next page*

*Figure 2 continued*

images of STF reporter fluorescence for indicated conditions. Scale bar 100 μm. (**C**) Quantification of STF mCherry reporter fluorescence in HFE cells co-cultured with AGS (n per condition = 322, 394, 258, and 275). (**D**) Relative number of HFE cells per image after co-culture with AGS cells expressing indicated constructs. Significance calculated by Student's t-test with Bonferroni correction for multiple comparisons. (n per condition = 28, 26, 27, and 17; n=number of images). (**E**) Representative images of proliferating, BrdU-stained (red); co-cultured AGS and HFE-145 cells, as described in (**a**). Cells were counterstained with haematoxylin (blue dots). Scale bar 100 μm. Complementary images show BrdU$^+$ cells with red dots; blue dots mark BrdU$^-$ cells. (**F**) Quantification of BrdU-stained cells as a percentage of the population. Significance calculated by Student's t-test with Bonferroni correction for multiple comparisons. (n per condition = 20, 20, 20, and 20; n=number of images). (**G**), Colony-forming assay of AGS cells. AGS cells were transfected with the indicated constructs and co-cultured with AGS-RFP cells for 2 days. After sorting, AGS-RFP expressing cells were plated at clonal density for 10–12 days; (**H**), Quantification of spherical colonies. Significance is calculated by Student's t-test (n=9).

The online version of this article includes the following source data and figure supplement(s) for figure 2:

**Source data 1.** Western blot images of HFE, AGS, MKN7, and MKN28 cell lysates stained for Flot2.

**Figure supplement 1.** Wnt3 cytonemes depend on IRSp53 function.

**Figure supplement 1—source data 1.** Western blot image of AGS cell lysates stained for Flot2.

which enhances the generation of filopodia-like structures in various cancer cell lines (*Neumann-Giesen et al., 2004*) and is highly expressed in GC (*Cao et al., 2013*; *Zhu et al., 2013*). We found that AGS cells express 2.1-fold Flot2 mRNA and 2.7-fold higher levels of Flot2 protein than HFE-145 (*Figure 3a*). Of all the GC cell lines, AGS cells express the most elevated Flot2 mRNA and protein levels (*Figure 3—figure supplement 1A*), which coincides with an increased number of filopodia (*Figure 1B*). We next asked whether Flot2 is involved in the generation of filopodia in gastric epithelial cells as well as in GC cells. We found that Flot2 can induce a significant increase in the number and length of filopodia in HFE-145, similar to the ones observed in AGS cells (*Figure 3B–D*, *Figure 3—figure supplement 1B*). Expression of Flot2 in AGS does not further increase the average filopodia length due to promoting an increase in both longer and shorter filopodia, which averages a similar length to wildtype (WT). Next, we blocked the Flot2 function by using the dominant-negative Flot2 mutant ΔN-Flot2-GFP, which lacks its N-terminus and cannot localise to the membrane (*Neumann-Giesen et al., 2004*). In addition to its inhibitory function, ΔN-Flot2-GFP removes endogenous Flot2 from the plasma membrane (*Figure 3—figure supplement 1C*). We also used a siRNA-mediated knock-down of Flot2 (*Figure 3—figure supplement 1D*). Consistently, transfection of ΔN-Flot2-GFP or Flot2 siRNA significantly decreased the filopodia length and number in both cell types (*Figure 3B–D*). Control AGS cells, as well as AGS and HFE-145 cells transfected with Flot2, display about 20% of filopodia with a length over 10 μm (*Figure 3E*). Concordantly, a reduction of Flot2 function led to an increase of short filopodia with a maximum length of 4 μm. A Pearson's $\chi 2$ test confirmed significance in the distribution of filopodia lengths following Flot2 manipulation. Additionally, cells with perturbed Flot2 function have more lamellipodia, suggesting a change in actin cytoskeletal dynamics (*Figure 3—figure supplement 1E*). Thus, we conclude that Flot2 enhances the formation and elongation of filopodia and thus might explain the differences of the cytoneme phenotype in the Flot2 low-expressing HFE-145 cells versus the GC cells displaying high levels of Flot2 expression.

## Flot2 co-localises with Wnt3 on cytonemes

Next, we investigated if Flot2 influences Wnt3 cytonemes rather than just filopodia. Therefore, we examined whether Flot2 and Wnt3 are co-localising on cytonemes. In AGS cells, we found that Flot2 displays punctate staining at the plasma membrane with noticeable staining on filopodia (*Figure 4A*). Next, we investigated whether these filopodia contain both, Wnt3, and Flot2. After double IHC analysis, we found that Flot2 and Wnt3 are co-localising on Wnt3 cytonemes (*Figure 4B*, *Figure 4—figure supplement 1A*). Similar to endogenous Flot2, Flot2-GFP localised along the length and to tips of filopodia (*Figure 4C*). Intriguingly, we detected that Flot2-GFP co-localises with Wnt3-mCherry on cytonemes (*Figure 4D*), similar to the endogenous protein. We further observed that Flot2-GFP and Wnt3-mCherry co-localise at the cytoneme contact sites and that Wnt3-mCherry localises in Flot2-GFP-positive vesicles intracellularly (*Figure 4D*, *Figure 4—figure supplement 1B*). Whether Flot2

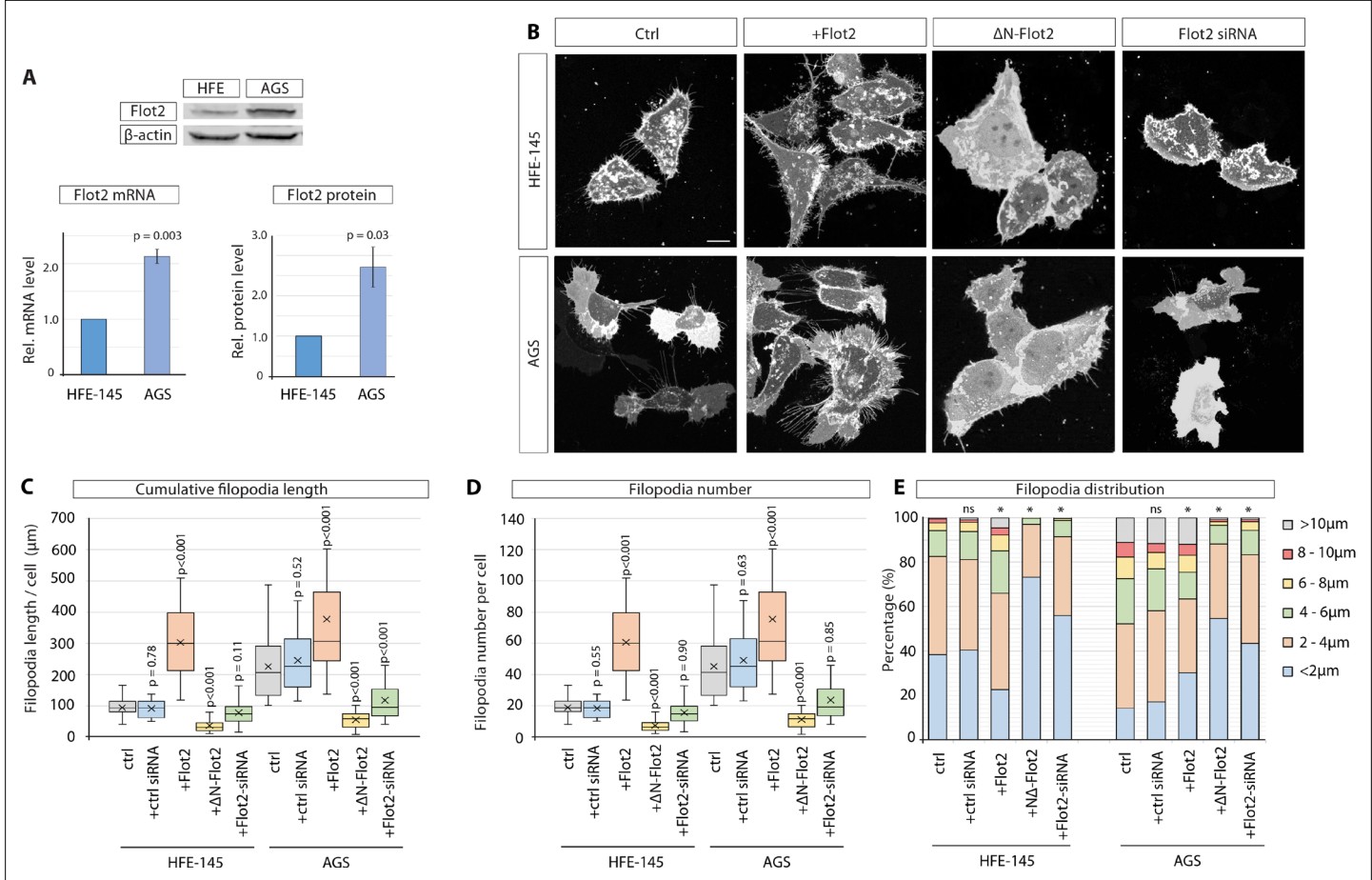

**Figure 3.** Flotillin-2 is over-expressed and promotes filopodia formation and elongation in gastric cancer cells. (**A**) Flot2 protein levels in HFE-145 and AGS cells as quantified by Western blot after normalising to beta-actin levels (n=3) and by RT-qPCR after normalising to Glyceraldehyde-3-Phosphate Dehydrogenase (GAPDH) as a housekeeping gene (n=4). Relative protein and mRNA levels are compared to HFE-145. Error bars represent SEM. Significance is calculated by Student's t-test. (**B**) Representative images of HFE and AGS cells expressing membrane-mCherry and indicated Flotillin-2 (Flot2) constructs or siRNA after 48 hr. Scale bars 10 μm. (**C–D**) Filopodia quantifications of HFE and AGS cells transfected with indicated Flot2 plasmids or siRNA. Significance calculated by Student's t-test with Bonferroni correction for multiple comparisons. Average cumulative filopodia length (**C**), average filopodia number per cell (**D**). (n per condition [HFE]=22, 19, 25, 23, 24). (n per condition [AGS]=25, 21, 25, 25, 25; n=number of cells measured). (**E**) Distribution of filopodia, categorised by length as a percentage of total filopodia per HFE or AGS cell 48 hr post-transfection with indicated Flot2 plasmids or siRNA. A Pearson's $\chi 2$ test was performed to test for significance between control (ctrl) group (expected) and experimental groups (observed) with 5 degrees of freedom (df) and a p-value <0.05. The specific $\chi 2$ values are as follows, HFE: ctrl siRNA 0.86, Flot2 0.001, dnFlot2 <0.001, Flot2 siRNA <0.001, and for AGS: ctrl siRNA 0.65, Flot2 0.007, dnFlot2 <0.001, and Flot2 siRNA <0.001. Asterisks mark significant differences.

The online version of this article includes the following source data and figure supplement(s) for figure 3:

**Source data 1.** Western blot images of HFE, AGS, MKN7, and MKN28 cell lysates stained for Flot2.

**Source data 2.** Western blot images of HFE, AGS, MKN7, and MKN28 cell lysates stained for Flot2.

**Figure supplement 1.** Effects of Flot2 on filopodia and Wnt3.

**Figure supplement 1—source data 1.** Flotillin-2 promotes filopodia formation.

**Figure supplement 1—source data 2.** Flotillin-2 promotes filopodia elongation.

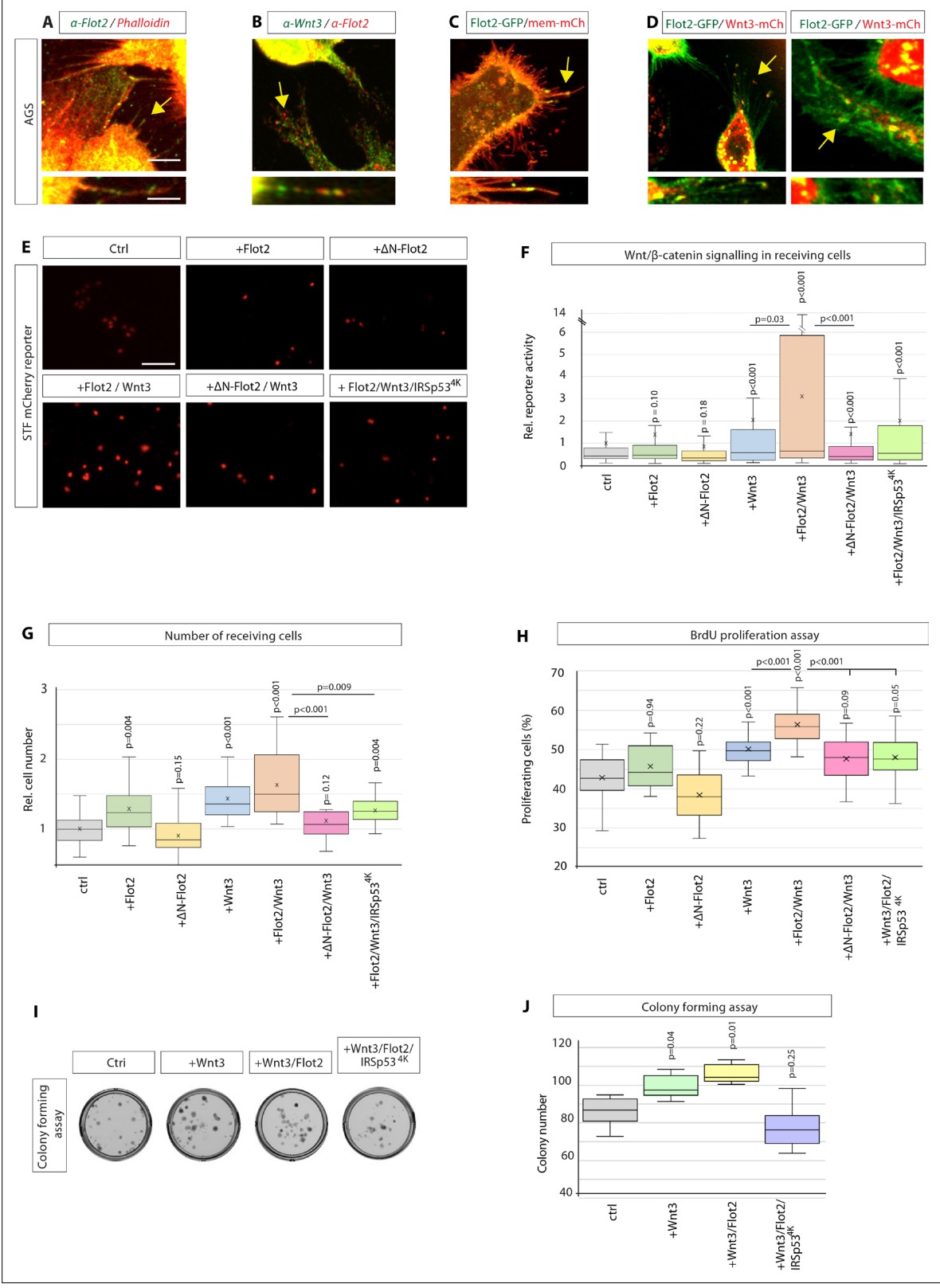

**Figure 4.** Flotillin-2 marks Wnt3 cytonemes and influences paracrine Wnt/β-catenin signalling and proliferation. (**A**) Immunohistochemistry (IHC) analysis showing endogenous localisation of Flot2 (green) in AGS cells. TRITC phalloidin was used to visualise actin. Arrows indicate the localisation of Flot2 to filopodia. Scale bars 5 µm. High-magnification images indicate an example of a Flot2-bearing cytonemes. Scale bars 2.5 µm. (**B**), IHC analysis shows that Flot2 co-localises with Wnt3 on cytonemes. (**C**) Confocal images showing the subcellular localisation of Flot2-GFP in AGS cells. Arrows indicate

*Figure 4 continued on next page*

*Figure 4 continued*

the localisation of Flot2-GFP on cytonemes. (**D**) Confocal images highlighting co-localisation of Flot2-GFP and Wnt3-mCh on cytonemes in AGS cells (arrows). Flot2-GFP and Wnt3-mCherry also cluster and co-localising at a cytoneme contact point (arrow). (**E**) Representative images of SuperTOPFlash (STF) reporter fluorescence for indicated conditions. Scale bar 100 uM. (**F**) Relative quantification of STF mCherry reporter fluorescence in HFE cells co-cultivated with AGS cells expressing indicated constructs. Quantifications are relative to AGS control. (n per condition = 322, 443, 403, 258, 336, 306, and 297; n=number of nuclei measured). (**G**) Relative number of HFE cells per image after co-cultivation with AGS cells expressing indicated constructs. Significance calculated by Student's t-test with Bonferroni correction for multiple comparisons. (n per condition = 28, 27, 26, 27, 22, 24, and 15; n=number of images). (**H**) Quantification of BrdU-stained cells as a percentage of the population, after co-cultivation of AGS and HFE cells, as described in *Figure 2a*. significance calculated by Student's t-test with Bonferroni correction for multiple comparisons. (n per condition = 20; n=number of images). (**I**), Colony-forming assay of AGS cells. AGS cells were transfected with the indicated constructs and co-cultivated with AGS-RFP cells for 2 days. After sorting, AGS-RFP expressing cells were plated at clonal density; (**J**) Quantification of colonies after 10–12 days. Significance is calculated by Student's t-test (n=9).

The online version of this article includes the following source data and figure supplement(s) for figure 4:

**Source data 1.** Flotillin-2 regulates paracrine Wnt/β-catenin signalling and proliferation.

**Figure supplement 1.** Co-localisation study of Flot2 and Wnt3.

**Figure supplement 2.** Colony formation assay.

and Wnt3 directly interact as part of a complex requires further research. However, from this data, we hypothesised that Flot2 is a marker of Wnt3 cytonemes in GC cells.

## Flot2 enhances paracrine Wnt/β-catenin signal activation and cell proliferation

To investigate a functional relationship between Wnt3 and Flot2, we assessed whether Flot2 function impacts paracrine Wnt signalling via cytonemes. First, we performed a co-cultivation assay of AGS cells and HFE-145 cells as done previously (*Figure 2A*). We observed that the expression of Flot2 in the AGS cells slightly increased the expression of the STF mCherry reporter in the neighbouring HFE-145 cells (*Figure 4E and F*). Consistently, ΔN-Flot2-GFP reduced paracrine Wnt signal activation in HFE-145. Strikingly, co-expression of Flot2 and Wnt3 significantly enhanced the STF reporter expression and, thus, paracrine Wnt signal activation. This increase is reduced upon coexpression of Wnt3 with ΔN-Flot2-GFP. Even more striking, the expression of the cytoneme antagonist IRSp53[4K] led to a significant reduction of paracrine Wnt signalling activation in Wnt3 and Flot2 expressing cells. We could not find changes in Wnt3 expression or subcellular localisation in cells expressing ΔN-Flot2 or Flot2 siRNA (*Figure 4—figure supplement 1C, D*). Thus, our results suggest that Flot2 significantly increases the number of Wnt3 cytonemes (*Figures 3 and 4*) and this increase led to a massive upregulation of paracrine Wnt signal activation (*Figure 4E and F*).

Next, we determined the impact of these changes in Wnt signal activation on cell numbers in Wnt-receiving HFE-145 cells (*Figure 4G*). We found that Flot2 expression caused a significant increase in HFE-145 cell number, while ΔN-Flot2 caused a decrease. Following a similar trend to the STF reporter, coexpression of Flot2 with Wnt3 resulted in the greatest increase, whereas the Wnt3-induced proliferation was attenuated by coexpression with ΔN-Flot2-GFP. Similarly, IRSp53[4K] attenuated cell number increases in the presence of Wnt3/Flot2, suggesting Flot2 significantly impacts proliferation in a cytoneme-dependent manner (*Figure 4G*). A BrdU-based cell proliferation assay confirmed the effects of Flot2-Wnt3 signalling on proliferation. We found that coexpression of Flot2 and Wnt3 produced the greatest increase in BrdU incorporation into newly synthesised DNA of actively proliferating cells. (*Figure 4H*). Consistently, IRSp53[4K] reduced BrdU labelling and thus proliferation. We cannot rule out that the modest inhibition of paracrine Wnt signal activation and proliferation by ΔN-Flot2-GFP at endogenous Wnt3 levels may be due to Wnt3 transport via a Flot2-independent mechanism.

We further tested the requirement for Flot2-controlled cytonemes in Wnt3 spreading in a CF assay (*Figure 4I*, *Figure 4—figure supplement 2*). After reverse transfection of AGS with Wnt3, Flot2, and IRSp53[4K], we performed a paracrine CF assay as described previously (*Figure 2*). The FACS isolated Wnt3 receiving GC cells were used for CF over the following 8–10 days. Quantification of the colonies revealed a significant increase in stemness of AGS cells when prior co-cultivated with Wnt3-expressing AGS cells (*Figure 4I and J*). To test for the requirement for Flot2 in cytoneme-mediated Wnt3 transport, we co-transfected the Wnt3-producing cells with Flot2. Indeed, we find that Wnt3/Flot2 expression increases the number of colonies and thus promotes stemness even further. Consistently, blockage

of cytoneme formation via co-expression of IRSp53[4K] reduced the number of colonies. Congruently, blockage of Flot2 function by expressing dominant-negative Flot2 (ΔN-Flot2-GFP) reduced the number of colonies (*Figure 4—figure supplement 2A, B*). These data suggest that Flot2-dependent cytonemes are vital regulators of paracrine transport for Wnt3 signalling, controlling the activity of GC stem cells.

## Flot2 influences intracellular transport and membrane localisation of Ror2

Next, we addressed the underlying mechanism for Flot2-mediated cytoneme formation. Previously, the Wnt co-receptor Ror2 has been shown to be critical for Wnt-mediated cytoneme formation and signalling (*Mattes et al., 2018*; *Brunt et al., 2021*). Since Ror2 has been shown to localise lipid rafts (*Sammar et al., 2009*), we hypothesised that Flot2 could interact with Ror2 to promote cytoneme formation. To test this hypothesis, we analysed first the intracellular distribution and found colocalisation of Ror2 and Flot2, particularly at clusters at the plasma membrane (*Figure 5A*), most likely Flot2 positive microdomains. Overexpression of tagged versions of Flot2 and Ror2 led to a similar colocalisation at the membrane and, consequently, the formation of long cytonemes (*Figure 5B*, *Figure 5— figure supplement 1A*). Strikingly, we found that siRNA-mediated knock-down of Flot2 resulted in a substantial reduction in Ror2 membrane localisation and its intracellular accumulation (*Figure 5C*). Consistently, we found that inhibiting Flot2 function (using ΔN-Flot2-GFP) also led to a loss of Ror2 membrane localisation and, subsequently, Ror2 accumulated in the perinuclear region (*Figure 5D*). These experiments suggest that Ror2 requires Flot2 for proper membrane localisation. To further assess the change in subcellular localisation, we mapped the localisation of Ror2 with regard to a set of organelle markers and found that upon siRNA-mediated Flot2 knock-down, Ror2-mCherry localisation significantly increases in the Golgi apparatus without a noticeable change in the other organelles investigated (*Figure 5E*; *Figure 5—figure supplement 1B-E*). This data set suggests Flot2 is required for Ror2 Golgi exit and trafficking between the Golgi and the membrane. Mechanistically, how Flot2 achieves this function requires further molecular investigation.

## Flot2 regulates Ror2/PCP signalling

Based on the observation that Flot2 is required for proper membrane localisation of Ror2, we hypothesised that Ror2 mislocalisation would also impact Ror2-mediated cytoneme formation (*Mattes et al., 2018*). During cytoneme outgrowth, Ror2 needs to activate the Wnt/PCP pathway, including downstream activation of JNK signalling, to induce polymerisation of the cytoneme actin cytoskeleton (*Martinez et al., 2015*; *Brunt et al., 2021*). Since Flot2 is required for Ror2 membrane localisation, we investigated if Flot2 function can be linked to Ror2/PCP signalling. Therefore, we generated an AGS cell line stably expressing a reporter of JNK signalling, the JNK kinase translocation reporter (JNK KTR-mCherry) (*Regot et al., 2014*; *Miura et al., 2018*; *Brunt et al., 2021*). JNK-KTR-mCherry localises to the nucleus (N) in its dephosphorylated state (low JNK activity). Upon activation by phosphorylation, it shuttles to the cytoplasm (C, high JNK activity). The C:N ratio can then be calculated as an indicator of JNK signalling. AGS cells display low basal levels of JNK activity, indicative of low basal Wnt/PCP signalling (*Figure 5g and h*). Expression of either Flot2 or Ror2 significantly increases JNK signalling, and co-expression of Flot2 /Ror2 further enhances JNK signalling activation, suggesting a synergistic effect. In accordance with a synergistic function, we demonstrated that blockage of Flot2 function in Ror2 positive AGS cells by co-expression of a ΔN-Flot2 can significantly reduce PCP/JNK signalling. This data set suggests that Flot2 is required for Ror2-induced JNK signalling. In parallel, we observed that Flot2 represses the autocrine β-catenin signalling pathway, which is consistent with previous observations suggesting a mutually repressive nature of the Wnt/PCP and Wnt/β-catenin pathways (*Figure 5—figure supplement 1F, G*; *van Amerongen and Nusse, 2009*; *Niehrs, 2012*). Together, these data imply that Flot2 is required for Ror2-induced JNK signalling but inhibits canonical signalling in the Wnt-producing cell.

## Ror2 and Flot2 are required for cytoneme formation

Finally, we wanted to decipher the combined role of Flot2 together with Ror2 in cytoneme induction. For this reason, AGS cells were transfected with WT or mutant Flot2 and Ror2 constructs. Membrane-mCherry expression was used to visualise and quantify cytonemes (*Figure 5I–K*). Expression of Ror2

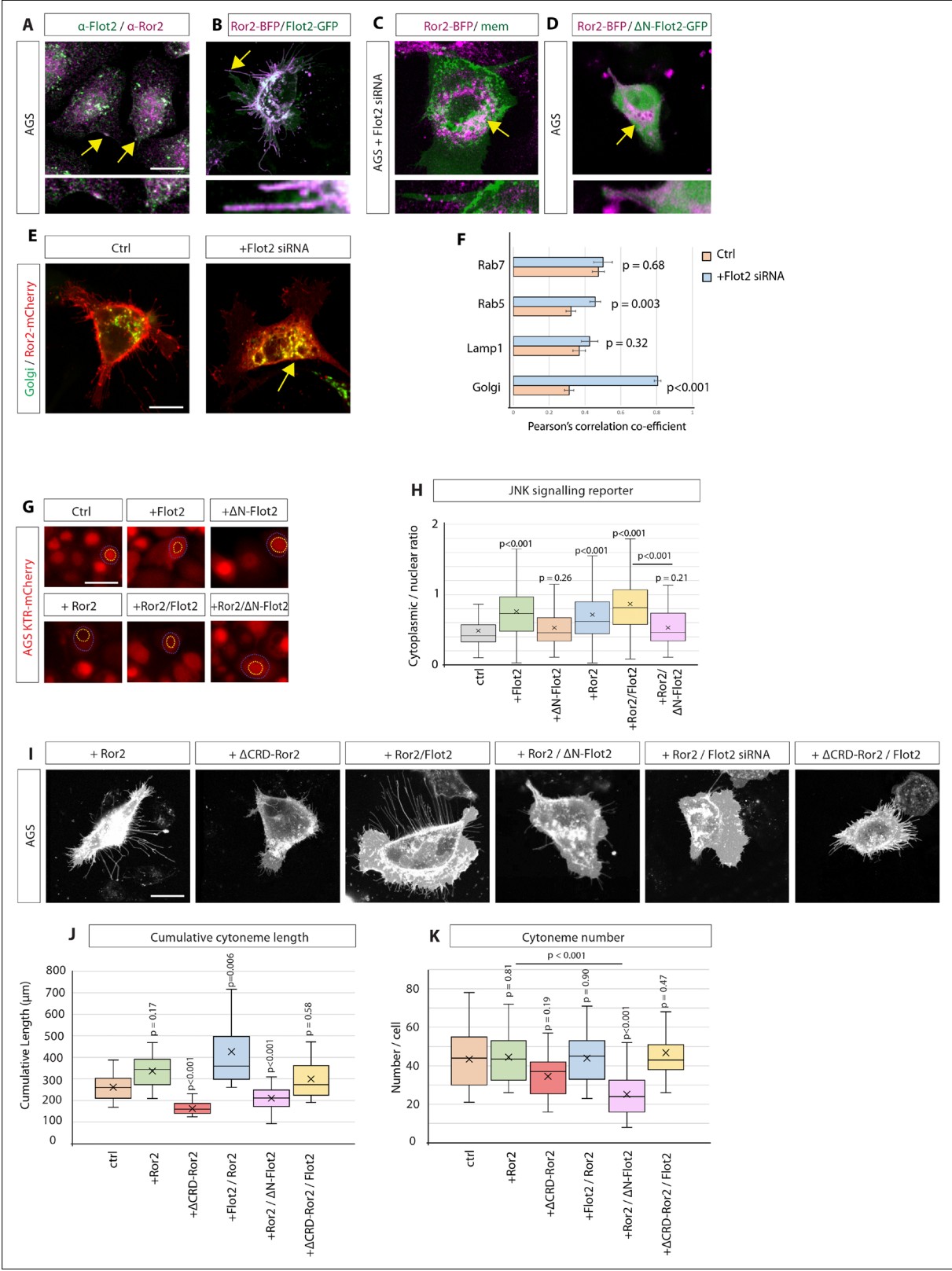

**Figure 5.** Flotillin-2 is required for Ror2 membrane localisation, Ror2/PCP signalling and Ror2-mediated cytoneme formation. (**A**) Immunohistochemistry (IHC) analysis of AGS cells stained for Ror2 (red) and Flot2 (green). Flot2 and Ror2 show co-localisation with a Pearson's correlation coefficient (PCC) of 0.65 (n=10), highlighted at the membrane by arrows. Scale bars represent 10 μm, and in high-magnification images, 2.5 μm (right). (**B–D**), Confocal live-cell imaging of AGS cells expressing Ror2-BFP with Flot2-GFP (**B**), Flot2 siRNA (**c**) or ΔN-Flot2-GFP (**D**) and memCherry. Arrows highlight subcellular

*Figure 5 continued on next page*

*Figure 5 continued*

regions of co-localisations. (**E**) Live confocal images of AGS cells expressing Ror2-mCherry and indicated organelle markers +/- Flot2 siRNA. Arrows highlight the co-localisation of Ror2-mCherry and mTurq2-Golgi. Scale bar 10 µm F, Quantification of co-localisation of Ror2-mCherry with indicated markers, assessed by PCC. Significance is calculated by Student's t-test. (n per condition [WT]=7, 10, 8, and 10) (n per condition [Flot2 siRNA]=7, 6, 7, 8). (**G**) Representative images of AGS cells stably expressing the JNK-KTR-mCherry reporter and indicated constructs after 48 hr. Blue dotted line encircles the cytoplasm and yellow dotted line the nucleus of a representative cell. Scale bar 20 µm. (**H**) Quantification of the JNK-KTR-mCherry reporter. Nuclear and cytoplasmic fluorescence of cells were measured, and the cytoplasmic: nuclear ratio was calculated. Significance is calculated by one-way ANOVA with Bonferroni correction for multiple comparisons. (n per condition = 136, 109, 74, 109, 82, and 79). (**I**) Representative confocal images of AGS cells expressing memCherry and indicated constructs for 48 hr. Scale bars 10 µm. (**J, K**) Quantification of cytoneme length and number of AGS cells transfected with constructs indicated in (**i**). Significance calculated by Student's t-test with Bonferroni correction for multiple comparisons. (n per condition = 25, 22, 21, 23, 25, and 21; n=number of cells).

The online version of this article includes the following source data and figure supplement(s) for figure 5:

**Source data 1.** Flotillin-2 is required for Ror2-mediated cytoneme formation.

**Figure supplement 1.** Flotillin-2 and Ror2 – localisation and signalling.

alone increased the average cytoneme length while having no significant effect on cytoneme number (*Figure 5J and K*). However, co-expression of Flot2 with Ror2 enhanced this phenotype; increasing cytoneme length. Consistently, blockage of Flot2 function upon expression of the dominant-negative ΔN-Flot2-GFP decreased the number and length of cytonemes in Ror2 expressing AGS cells (*Figure 5J and K*). To test whether Flot2 requires Ror2 for cytoneme formation, a mutant construct of Ror2 missing the Wnt-interacting CRD domain (ΔCRD-Ror2) was used, which cannot bind to Wnt ligands and thus transduce a Wnt signal. When expressed in AGS cells, ΔCRD-Ror2 significantly reduced the average length and number of cytonemes. We found that this phenotype can be partially rescued by the expression of Flot2, which moderately restores the average cytoneme length and number, suggesting that Flot2 acts downstream of Ror2-inducing cytonemes, potentially via its ability to enhance actin polymerisation. Taken together, these results show that Flot2 is required for the membrane localisation of Ror2. Furthermore, Flot2 facilitates Ror2-mediated Wnt/PCP signalling and, consequently, the promotion of Wnt cytonemes.

## Wnt8a cytonemes in zebrafish development require Flot2 function

To evaluate whether the role of Flot2 in forming Wnt cytonemes is conserved, we addressed Flotillin function during zebrafish development. The Flotillins are highly expressed in early zebrafish development and can be visualised on the tips of cellular protrusions (*Von Philipsborn et al., 2005*). Previous work from our lab has shown that cytonemes are essential for the dissemination of Wnt8a in zebrafish embryogenesis (*Stanganello et al., 2015*) and that these cytonemes can be regulated by the Ror2/PCP pathway (*Mattes et al., 2018*; *Brunt et al., 2021*). To map the subcellular localisation of Flot2 in zebrafish epiblast cells, we generated small cell clones expressing a fluorescence tagged Flot2 during zebrafish gastrulation and analysed the expression of Flot2 at 6hpf (*Figure 6A*). We found that human Flot2-GFP displayed strong membrane localisation and was localised to filopodia, including their tips (*Figure 6B*). Flot2-GFP could also be seen clustering together with Wnt8a-mCherry on cytonemes (*Figure 6C*), analogous to the co-localisation seen with Wnt3 in GC cells. Next, we altered Flot2 function during zebrafish embryogenesis and observed that the expression of Flot2 significantly increased the average cytoneme length and cumulative length while having no significant effect on cytoneme number (*Figure 6D–F*). Consistently, the dominant-negative ΔN-Flot2-GFP attenuated the cytoneme number and cumulative length. During blastula stages, Flot1b/Reg2b and Flot2a/Reg1a are predominantly expressed in the zebrafish (*Von Philipsborn et al., 2005*). Therefore, we used our CRISPR/Cas9- KO strategy in zebrafish embryos (*Winter et al., 2022*) to generate F0 knockout zebrafish embryos (Crispants) for Flot1b/2 a. (*Figure 6G*). We found an effective somatic mutation in the Crispants (*Figure 6—figure supplement 1*). Within this double Crispant background, we generated small cell clones expressing Wnt8a-GFP and mem-mCherrry (*Figure 6H*). We observed that the length and number of Wnt8a cytonemes are significantly reduced when Flot1b/Flot2a function is reduced (*Figure 6I*).

Next, we addressed the consequences of Flot2 function on forming the Wnt/β-catenin signalling gradient in zebrafish embryogenesis. Wnt8a is a key Wnt morphogen in determining the positions of the boundaries of the brain anlage, and alteration of Wnt8a cytonemes perturbs brain anlage

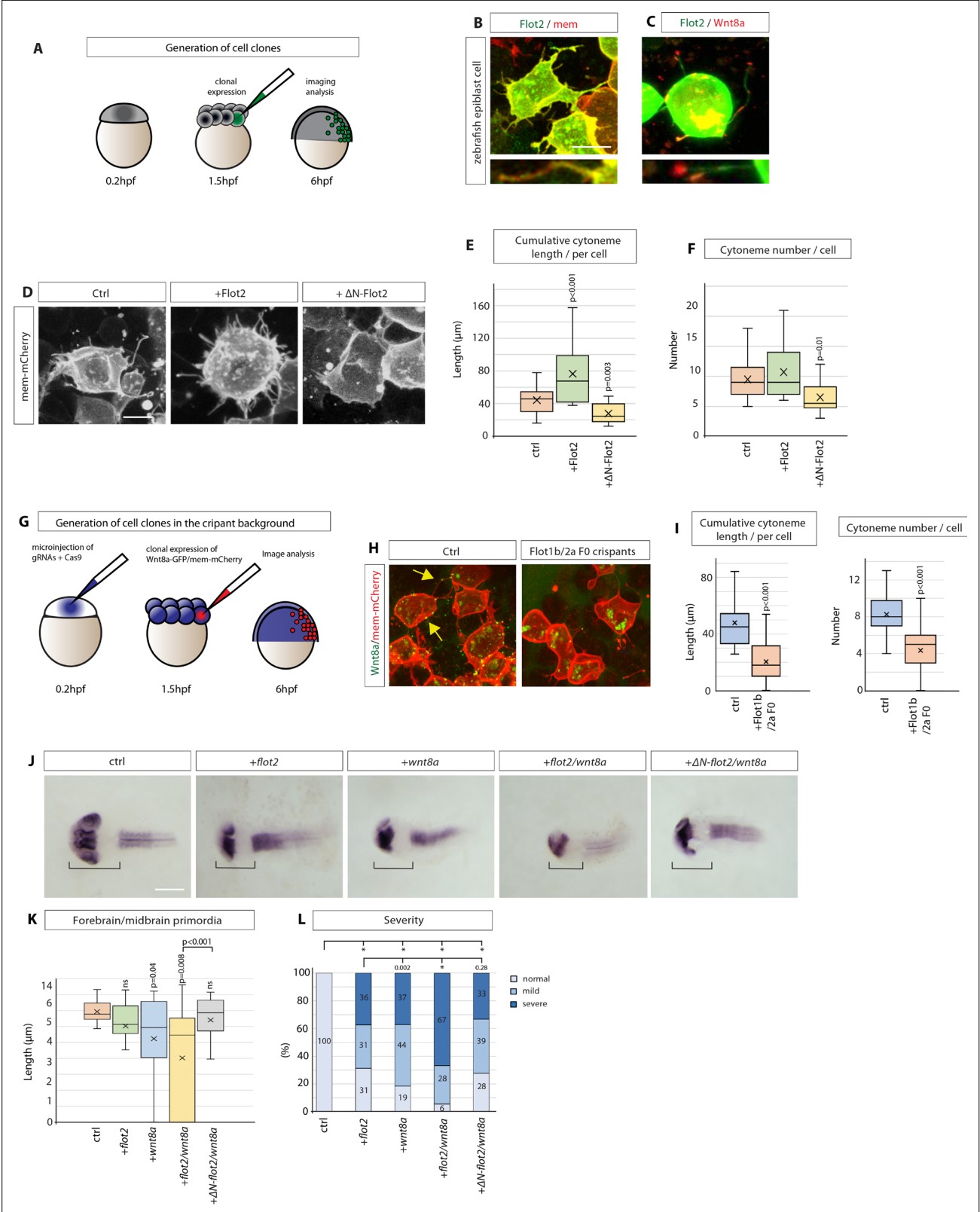

**Figure 6.** Flotillin-2 promotes cytoneme formation and Wnt8a signalling in zebrafish development. (**A**) Experimental setting to generate small clones expressing indicated constructs in the zebrafish embryo. (**B, C**) Confocal images of zebrafish epiblast cells injected with Flot2-GFP / memCherry and Flot2-GFP / Wnt8a-mCherry and imaged at 8 hpf. Scale bars represent 10 μm (**D**) Representative images of zebrafish epiblast cells injected with memCherry indicated constructs. Scale bar 10 μm. (**E**), (**F**), Quantification of filopodia from epiblast cells injected. Significance is calculated by

*Figure 6 continued on next page*

*Figure 6 continued*

Student's t-test. (n per condition = 17, 20, and 14). (**G**), Experimental strategy to generate Wnt8a-GFP/mCherry cell clones in F0 Flot1b/2 a Crispants background. (**H**), Confocal images of zebrafish epiblast cells expressing indicated constructs. (**I**) Quantification of cytoneme length and number in injected epiblast cells. Significance is calculated by Student's t-test. (n per condition = 18, 15) (**J**) In situ hybridisation against *pax6a* in zebrafish embryos at 30hpf after microinjection of 100 ng/µl of indicated DNA constructs and imaged. Scale bar represents 100 µm. (**K**) Quantification of forebrain and midbrain primordia length in zebrafish embryos injected as in (**A**). Significance is calculated by Student's t-test. (n per condition = 23, 16, 27, 18, and 18; n=number of embryos). (**L**) Qualitative analysis of phenotype severity in zebrafish embryos injected as indicated in (**A**). Phenotypes are classified into the categories normal, mild and severe. Numbers in bars represent percentages of total embryos. A Pearson's $\chi 2$ test revealed a significant difference between the ctrl group (expected) and experimental groups (observed) with 2 degrees of freedom (df) and a p-value < 0.05 of $\chi 2$ <0.001. Distribution comparison of *flot2* injected embryos revealed a significant difference to the *wnt8a* group ($\chi 2$=0.002) and the *flot2 + wnt8* a group ($\chi 2$<0.001), but not to *wnt8a+ΔN-flot2* ($\chi 2$=0.28). Asterisks mark significant differences.

The online version of this article includes the following source data and figure supplement(s) for figure 6:

**Source data 1.** Flotillin-2 promotes cytoneme formation in zebrafish development.

**Figure supplement 1.** Site-specific mutagenesis of zebrafish *flot2a* and *flot1b* genes.

patterning (*Mattes et al., 2018*; *Brunt et al., 2021*). Therefore, we altered Wnt expression levels together with Flot2 levels during neural plate patterning and found that the expression of low levels of Wnt8a results in an anterior shift of the boundaries of the brain anlage and reduces the combined forebrain (FB) and midbrain (MB) length (*Figure 6J and K*). Such a posteriorisation was also observed upon injection of Flot2-GFP. An even more pronounced shift of the position of brain anlage boundaries occurred upon co-injection of Flot2-GFP and Wnt8a, suggesting a synergistic effect of Flot2 and Wnt8a during posteriorisation of the neural plate patterning. In some cases, the anterior *pax6* positive expression domain, most likely the FB anlage, was completely lacking, which was recorded as '0'. Consistently, blockage of Flot2 function by expression of the dominant-negative ΔN-Flot2-GFP attenuated the alteration in patterning. As well as perturbing the length of the primordia of the FB/MB, alteration of Wnt8a signalling can cause defects in the early zebrafish embryo, which can include complete loss of the FB primordium and underdeveloped or missing eyes. We then categorised the developmental defects and found that embryos co-expressing Flot2 and Wnt8a significantly increased the number of embryos with severe defects (*Figure 6L*).

Together, these data suggest Flotillin can enhance the length and number of Wnt8a cytonemes and thus alter the Wnt8a signalling gradient, leading to a posteriorisation of the zebrafish brain anlage. These findings are concurrent with our results in GC cells showing Flot2 enhances paracrine Wnt signal activation. In both systems, Flot2 also co-localises with the Wnt ligands, Wnt3, and Wnt8a, and promotes cytoneme formation, which suggests a conserved role for Flotillin in promoting cytoneme-mediated Wnt dissemination in vertebrate tissue.

## Discussion

Wnt/β-catenin pathway activity, in addition to other stemness signals, is required to maintain the gastrointestinal crypt microenvironment (*Sato et al., 2011*). Wnt3 is an essential short-range signal at the crypt base (*Farin et al., 2012*), showing a strongly reduced diffusion capability (*Farin et al., 2016*). Similarly, our study of gastric epithelial cells indicates that Wnt3 remains membrane-tethered during transport and signalling (*Figure 1*). In support of these observations, experiments in *Drosophila* using an artificially membrane-tethered form of Wg showed that free diffusion is not necessary, as without the ability of free diffusion, *Drosophila* wing patterning was almost completely unaffected (*Alexandre et al., 2014*). Here, we provide evidence that gastric epithelial cells can load Wnt3 on cytonemes for intercellular distribution (*Figure 1*). In gastrointestinal adenomas and differentiated carcinomas, Wnt3 and Wnt cargo receptor Evi/Wntless are significantly increased and crucial for cancer cell proliferation and colony formation (*Voloshanenko et al., 2013*), and we show that Wnt3 and Evi/Wntless co-localise on cytonemes (*Figure 1*). Furthermore, the Wnt receptor Fzd7, which can bind Wnt3, transmits oncogenic Wnt signalling to drive the proliferation of gastric tumours in vivo (*Flanagan, 2019*). In conclusion, elevated levels of Wnt3 are essential for quickly proliferating gastrointestinal cancers; however, it is unclear how this hydrophobic ligand is disseminated within a tumour's context. In many cancer cells, the number of filopodia is increased (*Machesky and Li, 2010*), and LGR5[+] intestinal stem cells form long cytonemes (*Snyder et al., 2015*). Our data suggest that the cancer stem cell-like

AGS cells form a similarly high number of long filopodia, and we assign a fresh function to these filopodia (*Figure 1*). We propose that enhanced Wnt3 expression and an increase of cytonemes lead to augmented Wnt3 dissemination and signalling within GC cells, as well as increased cell proliferation and activity of cancer stem cells—a prerequisite for adenoma formation. In support of our hypothesis, in *Drosophila,* cytoneme-mediated FGFR signalling is similarly required for tumour growth and malignancy (*Fereres et al., 2019*). However, the underlying molecular mechanism increasing cytoneme emergence in cancers is unclear.

Flotillins are upregulated in many tumours (*Gauthier-Rouvière et al., 2020*; *Zhu et al., 2013*). Our data suggest that Flot2 expression correlates with enhanced cytoneme formation in GC. These results concur with previous studies demonstrating the ability of Flot2 to enhance filopodial phenotypes (*Hazarika et al., 1999*; *Neumann-Giesen et al., 2004*). We further show that increased Flot2-function leads to a further Wnt ligand dissemination in GC cells as well as in zebrafish embryogenesis. These observations are in concert with data in *Drosophila* showing that Flot2 promotes long-range Wg signalling in the wing imaginal disc (*Katanaev et al., 2008*). Consequently, blockage of Flot2 function reduces paracrine Wnt signalling significantly—despite the high levels of Wnt3 expressed in GC cells (*Figure 4*). Consequently, Flot2 function correlates positively with colony formation, suggesting that Flot2-dependent Wnt3 cytonemes are essential in promoting proliferation and activity of gastric cancer stem cells. We, therefore, speculate that GC cells require not only increased Wnt3 expression levels for tumour maintenance but that increased Flot2 expression allows more effective ligand dissemination in the tumour tissue.

Previously, Wnt/PCP signalling has been recognised as a key regulator of Wnt cytoneme emergence in vertebrates (*Mattes et al., 2018*; *Brunt et al., 2021*), where the receptor tyrosine kinase-like orphan receptor Ror2 has been implicated in activating this signalling pathway (*Ho et al., 2012*). Furthermore, autocrine Ror2/PCP signalling leads to the induction of Wnt cytonemes in zebrafish gastrula and mouse myofibroblasts (*Mattes et al., 2018*). Here, we provide evidence that Ror2 is similarly required for Wnt cytoneme formation in GC via the activation of Wnt/PCP/JNK signalling. However, we found that GC cells display more and longer filopodia despite sequencing data indicating that Ror2 expression is reduced in GC tissue (*Li et al., 2014*). We, therefore, suggest that the low levels of Ror2 seen in GC are functionally compensated by high levels of Flot2, which works synergistically to induce long cytonemes.

Surprisingly, we also found that Flot2 function is a prerequisite for Ror2-mediated Wnt/PCP signalling, as blocking Flot2 function reduces Ror2 membrane localisation, signalling, and cytoneme induction. In accordance with our findings, it has been suggested that Flot2 can influence receptor tyrosine kinase (RTK) signalling by providing a signalling microdomain at the plasma membrane (*Banning, 2014*). Furthermore, there is good evidence that Flotillins influence EGF, FGF, and MAPK signalling; however, Flotillins have contradictory effects depending on the experimental setting and the tissue context, and a unifying explanation is missing to date (*Amaddii et al., 2012*; *Banning, 2014*). Therefore, we hypothesise that Flot2—in addition to its function in organising microdomains at the membrane—is a critical regulator of intracellular protein transport. Thus, we suggest that the Flot2 function is a prerequisite for activating the Wnt/PCP-dependent cytonemes; however, further functional roles for Flot2 might exist and might also be modulated in a cell-type-specific manner.

As a mechanistic explanation, we suggest that Flot2 regulates intracellular trafficking of the Wnt/PCP co-receptor and RTK Ror2 from the Golgi apparatus to the plasma membrane for cytoneme induction. On the one hand, this is consistent with reports highlighting Flot2 as a regulator of membrane invagination and trafficking between endocytic compartments (*Frick et al., 2007*). In particular, it has been observed that cargo, including Flot2 itself, accumulates in the Golgi complex of HeLa, Jurkat, and PC12 cells when Flot2 function is obstructed (*Langhorst et al., 2008*). Consistently, we found that ectopic Flot2 expression leads to effective transport of Ror2 to the plasma membrane and, thus, to enhanced cytoneme formation in the source cells (*Figure 5*). Consequently, paracrine Wnt/β-catenin signalling is facilitated in neighbouring receiving cells. In accordance, we found that blockage of Flot2-induced cytonemes—without interfering with Wnt3 expression and secretion—can significantly reduce paracrine Wnt activation (*Figure 4*). This suggests that Wnt3 uses mainly Flot2/Ror2 dependent cytonemes for intercellular transport. Whether the canonical Wnt3 is a transient passenger on these cytonemes, directly binding to Ror2 or binding to Evi/Wntless, remains to be answered. However, there is indeed some evidence that Wnt3 can bind to Ror2 (*Billiard et al.,*

*2005*). Furthermore, Wnt8a is a further example of a Wnt/β-catenin ligand interacting with not only multiple frizzled receptors but also both Ror2 and Lrp6 co-receptors (*Mattes et al., 2018*). Clarifying whether Wnt3 displays similar capabilities will aid our molecular understanding of its cytoneme localisation.

However, Flot2 may not be limited to promoting Wnt transport; this function of Flot2 in cytoneme-mediated transport may represent a conserved role since Flot2 has previously been shown to enhance Hh cytoneme formation and gradients in *Drosophila* (*González-Méndez et al., 2017*). It would be interesting to investigate whether Hh and the receptors involved in Hh cytoneme formation, such as Ihog also localise to Flot2 microdomains and Flot2-dependent cytonemes, which would suggest a more general function for Flot2 in promoting morphogen transport.

It has been suggested that elevated Wnt/β-catenin signalling activity correlates with stem cell signatures in cancer (*Reya and Clevers, 2005*). Therefore, it is tempting to speculate that GC cells generate a 'niche'-like environment that allows them to maintain stemness. A crucial factor for maintaining the GC stem cell niche is elevated Wnt3 signalling. We show that GC stem cells can form a cytoneme-mediated network allowing a fast and precise exchange of Wnt3. As an underlying mechanism, we, therefore, suggest that Flot2 is crucial in modulating cytoneme emergence in the niche to facilitate long-range Wnt3 transport. Indeed, the expression levels of both Wnt3 and Flot2 correlate with the degree of intestinal tumour progression with poor prognosis (*Merlos-Suárez et al., 2011*; *Liu et al., 2018*). We propose that despite the presence of pathway-activating mutations in APC or β-catenin, the survival of gastrointestinal cancer cells remains dependent on Wnt ligand distribution on Flot2-controlled cytonemes. Thus, besides directly interfering with the Wnt signalling pathway, regulating Flot2-mediated Wnt cytonemes in the cancer stem cell niche could provide a novel strategy for combatting Wnt-related cancers.

## Materials and methods

### Plasmids and antibodies

The following plasmids were used in transfections and/or microinjections: pCS2 + membrane-mCherry (*Mattes et al., 2018*), pCAG-mGFP membrane-bound GFP (Addgene 14757), pEGFP-N1 Flot2-GFP (*Neumann-Giesen et al., 2004*), pEGFP-N1 ΔN-Flot2-GFP (*Neumann-Giesen et al., 2004*), 7×TRE SuperTOPFlash-NLS-mCherry (*Moro et al., 2012*), JNK KTR-mCherry (*Regot et al., 2014*), pCS2 + LifeAct GFP, pCS2 + Rab5 GFP (Jim Smith Group), Rab7-eGFP (from Rüdiger Rudolf), pEGFP-N1 LAMP1-mTurq2 (Addgene #98828), IRSp534K-mCherry/GFP (*Stanganello et al., 2015*), pCS2 + Wnt8a-mCherry (*Stanganello et al., 2015*), pCS2 + Ror2-mCherry (cloned in with ClaI and XbaI), pEGFP-N3 mRor2-ΔCRD-GFP (XhoI-XhoI mRor2 insert taken from pcDNA-mRor2, subcloned into SalI site of pEGFP-N3 vector), pCS2 + Ror2-eBFP2 (cloned by inserting eBFP2 into pCS2 + Ror2 plasmid using XbaI and SnaBI), pCS2 + Wnt3-mCherry (cloned from Addgene plasmid pcDNA3.2-Wnt3 (#35909) into pCS2±mCherry vector using ClaI and XbaI).

The following primary antibodies were used for immunofluorescence and/or Western blots: anti-Wnt3 (Abcam 116222), anti-Flotillin-2 (Abcam 96507), anti-Flotillin-2 (Abcam ab113661), anti-Ror2 (Santa Cruz H-1), anti-beta-actin (Proteintech 60008–1-Ig), anti-Myosin-X (Santa Cruz C-1), and anti-Evi (EMD Milipore YJ5). The following AlexaFluor (Thermofisher) secondary antibodies were used for immunofluorescence: goat anti-rabbit 488 (Abcam 150077), goat anti-rabbit 568 (Abcam 175471), donkey anti-goat 647 (Abcam ab150135). For Western blotting, goat anti-mouse IRDye800CW (Abcam 216772) and donkey anti-rabbit AlexaFluor680 (Abcam 175772) were used.

### Cell culture

The following cell lines, kindly authenticated and gifted from Dr. Toby Phesse (Cardiff University, UK), were used: gastric tubular adenocarcinoma cell line MKN7, derived from metastatic site (lymph nodes) and the gastric tubular adenocarcinoma cell line MKN28, derived from metastatic site (liver) and the primary gastric adenocarcinoma cell line AGS (*Flanagan, 2019*). As a control, the non-neoplastic epithelial cell line HFE-145, gifted from Dr. Hassan Ashktorab (Howard University, Washington, USA) was used. All cell lines were tested regularly for mycoplasma by endpoint PCR testing every 3 months and broth tests every 12 months.

AGS, MKN7, and MKN28 cells were maintained in RPMI-1640 (Sigma-Aldrich), and HFE-145 in DMEM (Thermofisher) media, all supplemented with 10% FBS and 1% Pen-Strep (Gibco). Cells were routinely passaged (with 0.05% EDTA-free trypsin [Thermofisher]) at ~80% confluency.

Transient transfections of cells, either reverse or forward, were performed using FuGeneHD (Promega) according to manufacturer's protocol (using 3:1 FuGene: DNA ratio). siRNA transfections were performed using Lipofectamine RNAiMAX (Thermofisher) according to manufacturer's protocol, with a final siRNA concentration of 30 pmol. Control siRNA (MISSION siRNA Universal Negative Control #1) and Flotillin-2 siRNA (Thermo Fisher 122408) were used.

## Zebrafish maintenance and husbandry

Wild Indian Karyotype (WIK) wild-type zebrafish (*Danio rerio*) were maintained at 28°C and on a 14 hr light/10 hr dark cycle (*Brand et al., 2002*). Zebrafish care and all experimental procedures were carried out in accordance with the European Communities Council Directive (2010/63/EU) and Animals Scientific Procedures Act (ASPA) 1986. In detail, adult zebrafish for breeding were kept and handled according to the ASPA animal care regulations and all embryo experiments were performed before 120 hr post fertilisation. Zebrafish experimental procedures were carried out under personal and project licenses granted by the UK Home Office under ASPA, and ethically approved by the Animal Welfare and Ethical Review Body at the University of Exeter.

## SuperTOPFlash assay

For autocrine signalling (single cell population), $1 \times 10^6$ HFE or AGS cells were reverse transfected with STF reporter plasmid (7×TRE-NLS-mCherry) along with indicated plasmids. Cells were incubated for 48 hr before image acquisition on a Leica DMI6000 SD with a 20 × objective.

For co-cultivation assays, $2 \times 10^6$ HFE-145 cells were reverse transfected with STF plasmid and $2 \times 10^6$ AGS cells with indicated plasmids in 6 well plates. After 24 hr incubation, both cell types were trypsinised, counted and $1 \times 10^6$ of each were co-cultivated in 6 well plates for a further 48 hr before image acquisition.

For all assays, at least 5 images were taken in random locations for each biological repeat on a 20 × objective. Fluorescence intensity of nuclei were measured on Fiji software. The number of fluorescent nuclei were also counted as a measure of proliferation.

## JNK KTR-mCherry assay

AGS cells stably expressing JNK-KTR-mCherry (AGS-JNK8) were reverse transfected with indicated plasmids and incubated for 48 hr. Cells were imaged on a Leica DMI6000 SD with a 20 × objective. At least 5 images were taken in random locations for each biological repeat. Fluorescence intensity of each cell's cytoplasm and nucleus was measured using Fiji software. After background subtraction, cytoplasmic to nuclear ratio (C:N) was calculated.

## Western blotting

Cell lysates were collected from cell pellets by resuspending in 100 uL of ice-cold TNT lysis buffer 20 mM Tris pH8.0, 200 mM NaCl, 0.5% Triton-X-100, 1 × cOmpleteTM EDTA-free protease inhibitor (Sigma Aldrich) per $1 \times 10^6$ cells. Cells were agitated in lysis buffer on ice for 30 min before centrifuging at 12,000 RPM, 4°C for 20 min and removing the supernatant. PierceTM BCA protein assay kit was then used (according to manufacturer's microplate protocol) to calculate protein concentrations.

For SDS-PAGE gel electrophoresis, required volume of 4 × Laemmli buffer (with 10% BME) was added to protein samples and incubated at 95°C for 5 min. 50 µg of protein was then loaded into BIORAD Mini-PROTEAN precast gels (12% acrylamide) and run at 60 V for 2 hr. Semi-dry transfer onto nitrocellulose or PVDF membrane was performed (Thermo Scientific Pierce G2 fast blotter) for 15 min at 15 V, 1.3 A. Membranes were then blocked in 5% non-fat milk for 1 hr at RT before adding primary antibodies to desired concentrations and incubating on a roller overnight at 4°C. The following day, membranes were washed 3 × with TBS-T before incubating in secondary antibodies in TBS-T for 1 hr at RT. Membranes were then washed 2 × with TBS-T and 3 × with TBS before imaging on LiCor Odyssey CLx and processing on ImageStudio software.

## RT-qPCR

RNA for qPCR was collected from cell pellets using QIAGEN RNeasy kit according to manufacturer's protocol. RT-qPCR was then performed using SensiFAST SYBR Lo-ROX one-step kit with half volumes

according to manufacturer's protocol and run using Applied Biosystems QuantStudio6 Flex. Primer sequences used were as follows:

*Flotillin-2* (Forward 5'-GGCTTGTGAGCAGTTTCTGG-3'); (Reverse 5'-AATCTGCTCCACTGTCA GGG-3'), *GAPDH* (Forward 5'-GTCTCCTCTGACTTCAACAGCG –3'); (Reverse 5'- ACCACCCTGTTG CTGTAGCCAA –3').

### Antibody staining and image acquisition

Cells were plated onto glass coverslips and following indicated treatment/incubation, cells were washed in 1×PBS and fixed using 4% paraformaldehyde (PFA) (10 mins, RT) or modified MEM-Fix (4% formaldehyde, 0.25–0.5% glutaraldehyde, 0.1 M Sorenson's phosphate buffer, pH7.4) (*Bodeen et al., 2017*; *Rogers and Scholpp, 2021*) for 7 min at 4°C. Cells were then incubated in permeabilisation solution (0.1% Triton-X-100, 5% serum, 0.1 M glycine in 1×PBS) for 1 hr at RT. Primary antibodies were diluted in incubation buffer (0.1% Tween20, 5% serum in 1×PBS) and coverslips incubated in 50 µl spots on parafilm overnight at 4°C. Coverslips were then washed with 1×PBS 3 × for 5 min before incubation in 50 µl spots of secondary antibodies diluted in incubation buffer for 1 hr at RT. Coverslips were then washed 3 × for 5 min with 1×PBS before mounting onto glass slides using ProLong Diamond Antifade Mountant (Invitrogen) and left to dry for 24 hr before imaging. Confocal microscopy for immunofluorescent antibody imaging was performed on an inverted Leica TCS SP8 X laser-scanning microscope using the 63 × water objectives.

### BrdU proliferation assay

AGS and HFE-145 cells were co-cultivated on glass cover slips in 6 well plates. After 45 hr co-cultivation, cells were incubated in media containing 10 µg/ml BrdU (Abcam, ab142567) for 3 hr. Cells were then fixed with 4% PFA for 15 min at RT. After washing 2 × with PBS, immunostaining of BrdU was performed using a BrdU immunohistochemistry kit (Abcam, ab125306) according to manufacturer's protocol. Cover slips were then mounted onto slides using Prolong Diamond Antifade Mountant and left to dry for 24 hr. Slides were then imaged on a light microscope with 20 × objective and captured using an Olympus EP50 colour camera, with at least 10 separate locations imaged per repeat. BrdU-stained cells (brown) were counted and quantified as a percentage of the total cell population (counterstained blue with haematoxylin).

### Colony formation assay

The CF was performed with RFP-labelled AGS (AGS-RFP) cells. These AGS-RFP cells were generated by transduction with pHIV-H2BmRFP lentiviral vector and purified by FACS (*Tornillo et al., 2018*). We termed this cell line 'receiver cells'. On the other hand, the AGS 'producer' cells were reverse transfected with the indicated plasmids. Twenty four hours after transfection, the AGS producers were co-cultured with AGS-RFP receivers (1:1) for 2 days. Next, AGS-RFP cells were sorted by FACS using an Aria (Becton Dickinson) flow cytometer. DAPI staining was used for dead cell exclusion. The AGS-RFP cells were then seeded for colony-forming assays (500 events/well in 12-well plates). After 8–10 days whole-well images were taken using the S3/SX1 Incucyte platform and number of resulting colonies was determined with the ImageJ software.

### Zebrafish microinjections and image analysis

For experiments in zebrafish embryos, indicated DNA plasmids were injected at the 1–2 cell stage at 100 ng/µl after dechorination. Embryos were left to develop at 28°C until 8 hpf (shield stage). For confocal microscopy analysis, live zebrafish embryos were embedded in 0.7% low melting agarose (Sigma-Aldrich) dissolved in 1 × Ringer's solution. Images of embryos were obtained with an upright Leica TCS SP8X microscope equipped with hybrid detectors (HyD) using 63 × dip-in objective.

### Generation of F0 crispant zebrafish embryos

For the zebrafish Flotillin1b and Flotillin2a gene, three individual guide RNAs (gRNAs) were designed using CHOPCHOP (https://chopchop.cbu.uib.no) to target discrete sections of coding exon 1 or 2. The sequences of gRNA target sites are as follows: *flot2a* (5'-GGTAGGACCGAACGAAGCCC<u>TGG</u>-3', 5'-ATGGGGAATTGCTACACGGT<u>AGG</u>-3' and 5'-GGTGATAAGCCACCATGCCC<u>AGG</u>-3'; PAM sequences are underlined), *flot1b* (5'-CCTCCTCTCATGATTGCTGG<u>TGG</u>-3', 5'-GGTGTCCCAATCT

CAGTGAC<u>TGG</u>-3' and 5'-TGAGATTGGGACACCGTGTC<u>TGG</u>-3'). For the precise locations of gRNAs see *Figure 6—figure supplement 1A*. When selecting gRNAs, we excluded candidate target sites that had potential off-target sites with less than 2 mismatch sequences in the genome, and also confirmed that off-target sites with 3 mismatch sequences were not located in any exon or UTR region of the protein-coding gene. Selected gRNAs were also BLAST-checked to confirm a lack of off-target gene interaction. Cas9 protein was considered likely to be the most biologically active component of the experimental injection mixture and, therefore, the most appropriate injection control for use when assessing multiple candidate genes.

Prior to injection, gene-specific crRNA and tracrRNA (Integrated DNA Technologies Inc Coralville, United States) were diluted to a final concentration of 12 µM in nuclease-free duplex buffer and the resultant gRNA mixture incubated at 95°C for 5 min (*Winter et al., 2022*). Immediately prior to use, 5 µl of the gRNA mixture was mixed with Cas9-NLS protein (final concentration of 5 µM. New England Biolabs, Ipswich, United States), 2 M KCl (final concentration of 300 mM), and 0.5% v/v Phenol red solution (Sigma Aldrich Ltd. Poole, United Kingdom) in a total volume of 10 µl. The resultant mixture was incubated at 37°C for 10 min to assemble the gRNA/Cas9 ribonucleoprotein complex, and then held at room temperature until use. The efficacy of the CRISPR/Cas9 mutagenesis in injected fish was tested by a T7E1 assay of the genomic DNA from individual larvae. Primer sequences used were as follows: *flot2a* (Forward 5'-GGTGCTGCATTCATACGAGG-3'; Reverse 5'-GATGAACATGTAACGGA CTGC-3'), *flot1b* (Forward 5'-AGGATCACTCTCAATACTCTG-3'; Reverse 5'-TTCTGACCCTGGATCTT CAC-3').

## In situ hybridisation (ISH)

Pax6a digoxigenin (DIG) antisense RNA probes were generated from linearised plasmids using an RNA labelling and detection kit (Roche) (*Scholpp and Brand, 2003*). Embryos for ISH were dechorinated and injected with indicated DNA plasmids (100 ng/µl) at the 1–2 cell stage. Embryos were left to develop at 28°C for 30 hr before fixation in 4% PFA overnight at 4°C. Embryos were then washed twice in PBST and dehydrated in 100% methanol for 30 min at RT. Following 2 × PBST wash, they were re-fixed in 4% PFA for 30 min and washed again 2 ×. Embryos were then incubated in Hyb+ solution at 69°C for 4–6 hr before replacing the solution with probe-Hyb+ mix (1:20 Pax6a probe in Hyb+) and incubated overnight. The next day, embryos underwent several washing steps in 25% Hyb-, Hyb-, 2×SSCT and 0.2×xSSCT solutions at 69°C, then MABT at RT. Embryos were then incubated in 2% blocking buffer for 4–5 hr, followed by 1:4000 anti-DIG antibody (Roche) in 2% blocking buffer overnight at 4°C. The next day, embryos were washed 5×15 min in MABT and then once in NTMT (5 min). Embryos were transferred to 24 well plates and NTMT-NCP-BCIP mix (1:200 dilution NCP-BCIP:NTMT [Roche]) was added. Embryos were left to develop signal staining for 2 hr in the dark (RT) prior to NTMT and PBST washes and 30 min refix in 4% PFA (RT). Two final washes in PBST were performed before storing embryos in 70% glycerol. Stained embryos were then imaged on a stereo microscope and measurements of the forebrain and midbrain made using Fiji software. Pearson's $\chi 2$ test was used to calculate whether the distribution of embryos into severity classes differed significantly between the control embryos and the microinjected constructs.

## Quantifications and statistical analyses

All filopodia quantifications were calculated from Z-stack images of cells expressing membrane-mCherry and were done using Fiji software. Filopodia length was measured from the tip of the filopodia to the base, where it contacted the main cell body. In the case of branching protrusions, one branch (the longest) would be measured. Quantifications of Western blot images was performed by measuring the mean grey value of bands (after subtracting background) and normalising to loading controls. Quantification of qPCR data used the Pfaffl equation to calculate fold-change expression from Ct values, after normalising to GAPDH. PCC quantifications were performed using the 'Coloc-2' plugin in Fiji. A region of interest was drawn around the desired cell and PCC was measured with no threshold. All experiments/conditions were repeated at least in biological triplicates. Significance was tested using Student's t-test (parametric) against the relative controls. Bonferroni correction for multiple comparisons was applied. Pearson's $\chi 2$ test (non-parametric) was used to calculate whether the distribution of filopodia lengths differed significantly between the control cell lines and the transfected constructs. Error bars on bar charts show standard error of the mean (SEM).

## Acknowledgements

The Medical Research Council (MRC)/UKRI funded this project through a studentship for DR (MR/N0137941/1 for the GW4 BIOMED DTP) awarded to the Universities of Bath, Bristol, Cardiff and Exeter, an MRC research grant (MR/S007970/1) and a BBSRC RM grant (BB/S016295/1) supporting SR, LB, and YO awarded to SS, and an MRC research grant (MR/R026424/1) supporting VM and GT awarded to TJP We would like to thank Ritva Tikkanen for generously providing the Flotillin-2 constructs used in this study. In addition, we would like to thank Sahin Deniz (Istanbul) and Duane T Smoot (Nashville) for generating the HFE-145 gastric epithelial cell line. Finally, we would also like to thank Chengting Zhang for technical help and the entire Scholpp lab for their comments on the manuscript.

## Additional information

### Funding

| Funder | Grant reference number | Author |
|---|---|---|
| Medical Research Council | MR/N0137941/1 | Daniel Routledge Steffen Scholpp |
| Medical Research Council | MR/S007970/1 | Steffen Scholpp |
| Biotechnology and Biological Sciences Research Council | BB/S016295/1 | Steffen Scholpp |
| Medical Research Council | MR/R026424/1 | Toby J Phesse |

The funders had no role in study design, data collection and interpretation, or the decision to submit the work for publication.

### Author contributions

Daniel Routledge, Conceptualization, Data curation, Formal analysis, Validation, Investigation, Visualization, Methodology, Project administration, Writing - review and editing; Sally Rogers, Conceptualization, Data curation, Investigation, Methodology; Yosuke Ono, Lucy Brunt, Valerie Meniel, Giusy Tornillo, Investigation; Hassan Ashktorab, Resources; Toby J Phesse, Conceptualization, Data curation; Steffen Scholpp, Conceptualization, Supervision, Funding acquisition, Validation, Writing - original draft, Project administration, Writing - review and editing

### Author ORCIDs

Daniel Routledge  http://orcid.org/0000-0002-8259-349X
Toby J Phesse  http://orcid.org/0000-0001-9568-4916
Steffen Scholpp  http://orcid.org/0000-0002-4903-9657

### Ethics

Zebrafish care and all experimental procedures were carried out in accordance with the European Communities Council Directive (2010/63/EU) and Animals Scientific Procedures Act (ASPA) 1986. Zebrafish experimental procedures were carried out under personal and project licenses granted by the UK Home Office under ASPA, and ethically approved by the Animal Welfare and Ethical Review Body at the University of Exeter.

### Decision letter and Author response

Decision letter https://doi.org/10.7554/eLife.77376.sa1
Author response https://doi.org/10.7554/eLife.77376.sa2

## Additional files

### Supplementary files

- Transparent reporting form

## Data availability

All data generated or analysed during this study are included in the manuscript, supporting files and source files; Supporting Data files and Source Data have been provided to all figures.

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
