## [Editor Report]

This work should be of interest to cell and developmental biologists in the Wnt and cytoneme fields. The authors convincingly demonstrate that the membrane tethered scaffolding protein Flotillin-2 localizes to and stimulates Wnt cytoneme growth in gastric cancer cells and Zebrafish. This study extends previous cytoneme studies and provides new details about conserved regulatory events controlling Wnt ligand distribution.

---

## [Decision Letter]

**Decision letter after peer review:**

Thank you for submitting your article "The scaffolding protein Flot2 regulates cytoneme-based transport of Wnt3 in gastric cancer" for consideration by *eLife*. Your article has been reviewed by 3 peer reviewers, and the evaluation has been overseen by a Reviewing Editor and Didier Stainier as the Senior Editor. The reviewers have opted to remain anonymous.

Essential revisions:

1) The issues identified throughout the manuscript by all three reviewers regarding statistical analyses, reagent validation and control experiments have to be addressed.

2) Careful validation of overexpression tools used and back up with alternative approaches is needed.

3) Data presentation and description, especially microscopic imaging, has to be improved.

*Reviewer #2 (Recommendations for the authors):*

Figure 1A: Only single cells are shown to represent each of the cell types assayed. Zoomed-out images are needed for the reader to gauge what extensions look like across the cell population. It is difficult to determine which extensions are considered cytonemes and how they differ from other observable filopodia.

Figure 1H-I: There are a lot of red dots that do not appear to be associated with cells or filopodia on cells. Is this a background or signal specificity issue?

Figure 2B: It is difficult to determine the relative positivity of Wnt signaling activity in the cells assayed because only signal-positive cells are visible. A DAPI co-stain or DIC image is needed to see how many cells are in the field. Same issue for Figure 4E.

Figure 2C: How is the relative reporter activity determined? Relative to what? Please provide this information in the results or figure legend.

Lines 99-100: The authors conclude the majority of Wnt3 protein is disseminated by IRSp53 dependent cytonemes in gastric cells. It is not clear this can be concluded from the data shown. Is there a way to specifically monitor cytoneme dependent vs. secretion-based Wnt reporter activation? Does the IRSp534K mutant have any effect on Wnt secretion into culture media?

The IRSp534K cells shown in Sup Figure 1A look very unhealthy, so might IRSp534K be having off-target effects – like cell death? The membrane blebs are concerning. Are cells similarly affected upon Cdc42 or N-Wasp inhibition or dominant-negative protein expression?

Figure 2D: How were HFE-145 cells specifically quantified? Are they labeled differently than the cocultured AGS cells? Please provide a brief explanation how sending vs receiving cells were identified and accounted for in the quantification. Same question for Figure 2E-F.

Line 117: Is cytonemal a word?

Figure 3B: The extensions in HFE-145 that are increased upon Flot2 expression look more cytoneme-like than the extensions in AGS cells. Many of the AGS cell extensions look like retraction fibers. How are cytonemes being distinguished from retraction fibers? Please indicate in the figure what is a cytoneme and what is not. Were only cytoneme-like extensions quantified in C-E?

Figure 4B: Line 143 states that the authors want to determine whether Flot2 is specific to Wnt cytonemes and not just filopodia, so they look for Wnt3 and Flot2 localizing to the same extensions. Line 147 states that Wnt3 and Flot2 colocalize on cytonemes, but the wording is vague. Is the suggestion that Wnt3 and Flot2 are colocalizing as a protein complex, or is it that they are both present in the same cytonemes but not necessarily in complex? If the former, please provide colocalization coefficients. Same question for 4D. Even if Wnt3 and Flot2 do colocalize, this will not confirm Flot2 is dedicated to Wnt cytonemes. To make this conclusion, the authors will need to determine whether Flot2 does or does not colocalize with other signaling proteins that are transported along cytonemes. The authors cannot rule out that Flot2 is a general modulator of cytoneme transport.

Figure 5A: Quantification of colocalization is needed. Figure 5B: Does the arrow indicate a Flot2 punctum in a cytoneme, or is it simply indicating the cytoneme? Figures 5C-D: Quantification of changes in cytoneme formation and images of more than one cell for each condition are needed to gauge the extent to which cellular extensions are impacted by Ror2 and Flot2 gain/loss.

Figure 5I: Please add arrows to indicate examples of what were counted as cytonemes vs. non-cytoneme cell extensions that are present.

*Reviewer #3 (Recommendations for the authors):*

1) Line 112: authors say "that co-expression of Wnt3 together with IRSp534k led to a significant downregulation of actively proliferating cells" however, quantification in fig2F appears to show a very weak effect and whether the difference between +Wnt3 and +Wnt3/IRSp534K is significant is not clear. The p-values comparing various treatments (as given in 2d) should also be given for 2f.

2) Line 143: The authors say that they investigated if Flot2 specifically regulates Wnt3 cytonemes. However, they only show that Flot2 and Wnt3 colocalize on cytonemes, which does not directly suggest that Flot2 perturbation will only affect Wnt3 containing cytonemes.

3) Figure 4: It is intriguing to see that treatment of the control AGS cells with ΔN-Flot2, did not completely abolish the Wnt signalling activity or cell numbers/BrDU incorporation; even though it led to a significant reduction in the cytoneme numbers and length (Figure 3). As the GC cells are supposed to have higher Wnt3, which is suggested to enhance their proliferation, is it possible that the lack of a strong phenotype could be because of alternative mechanisms of Wnt3 release which remained unaffected by ΔN-Flot2 treatment?

4) Figure 5C and d are quite confusing. The text in lines 189-190 says "Consistently, expression of a dominant-negative form of Flot2 resulted in a strong reduction of cytonemes (Figure 5c) " while Figure 5C is marked as Flot2 siRNA. Mentioned in line 180, "we observed a similar phenotype after knock-down of Flot2 – a reduction of cytonemes and removal of Ror2 from the membrane (Figure 5d)" while Fig5d is about DN-Flot2.

5) Line 283: I think authors should rephrase the statement about *Drosophila* Wg being not diffusible, as there are several studies that have shown the long-range movement of endogenous Wg via a collection of extracellular carriers. Whereas the function of membrane-tethered Wg by JP Vincent group was shown via an artificially made Wg fusion protein and not for endogenous Wg.

6) Line 339-341: With the statement "Consistently, we found that ectopic Flot2 expression leads to an effective transport from Ror2 to the plasma membrane and thus, to enhanced Wnt3 cytoneme formation in the source cells (Figure 5)" authors are implying that they found Ror2 to be the mediator of Wnt3 transport on cytonemes. However, the experiments analysing the direct effect of Ror2 manipulation on Wnt3 transport have not been shown in Figure 5 and this conclusion appears to be based on their interaction shown by other studies.

Other comments:

Fig3E: Y-axis label should be added.

Line170: "whereas the Wnt3-induced signalling was attenuated by co-expression with ΔN-Flot2-GFP". This sentence is confusing as the data in Figure 4g is about the cell number and not signalling readout.

Line 171: "Similarly, IRSp534L attenuated increases seen in the presence…" appears to be an incomplete sentence.

[Editors' note: further revisions were suggested prior to acceptance, as described below.]

Thank you for resubmitting your work entitled "The scaffolding protein Flot2 promotes cytoneme-based transport of Wnt3 in gastric cancer" for further consideration by *eLife*. Your revised article has been evaluated by Didier Stainier (Senior Editor) and a Reviewing Editor.

The manuscript has been greatly improved but there are a few remaining issues that need to be addressed:

1) Please address text changes recommended by reviewer 3.

2) Please address the open concerns of reviewer 1, as outlined below.

As two reviewers had difficulties finding and understanding changes referred to in supplementary figures, please make sure the remaining changes are highlighted and the labeling of figures is consistent within text and letter.

*Reviewer #1 (Recommendations for the authors):*

The authors have improved the manuscript significantly.

There are a few issues remain that should be addressed.

General issues:

1. The naming system of the supplementary figures is very confusing. Most supplementary figures are not labeled, and some figures were mislabeled. The J editorial support team suggested downloading each figure and renaming them according to the file title. Even by doing that, many figures cannot be located. It seems that supplementary figures are associated with the main figures. The supp Figure 1 for Figure 2 is named Figure 2 supp Figure 1. Two supplementary figures for Figure 4, designated as Figure 4 suppl Figure 1, and Figure 4 suppl Figure 2. These names were used in the main text but not in the response letter.

For example,

1) Supp. Figure 1E: seems to be Figure 3 suppl figure 1C.

2) Supp. Figure 1C: seems to be Figure Suppl Figure 1A

3) Supp. Figure 2D: I found this panel in Figure 3 suppl figure 1D.

4) Supp. Figure 2E: cannot be located.

5) Figure 6D, E; Supp. Figure 3d: Supp. Figure 3d was not found.

6) One supplemental figure is labeled as supplementary figure 6. Is it correct?

Please review the manuscript, label all figures, and cite figures properly. Otherwise, the manuscript is impossible to be understood.

2. The responses were extensive, but the authors did not describe or label where the changes were, which makes the evaluation very challenging.

Specific issues:

The following issues were not addressed adequately:

1) The authors did not address this issue: "Figure 2D shows Relative number of HFE cells per image".

Did the authors use the number from the control as the base? Is it more meaningful if the actual values are used?

Additionally, is it possible to image Wnt3 is being transported to the receiving cells?"

The authors did not address if a video is provided. The Wnt3 staining in Figure 2 Suppl Figure 1A is messy. Are all GFP staining Wnt3 signals? Where was Wnt3 expressed? Which cells are the recipient cells?

2) "Figure 2E: The BrdU stains are brightfield images and do not use any fluorescence. Therefore, there are no channels to split. "

Could authors show large images for better visualization if this is the case?

3) "we performed antibody staining against Wnt3 after Flot2 KD (and in DN-Flot2-expressing cells) to assess its localisation (Supp. Figure 2)."

This figure cannot be located. Is it Figure 3, suppl Figure 1C?

4) "For clarification, we have added time-lapse images of Flot2-GFP/Wnt3-mCh expressing cells (Supp. Figure 2G), where Flot2 and Wnt3 can be seen travelling together intracellularly."

Is it Figure 4, supp. Figure 2B? This figure cannot be located. Time-lapse was not provided. The still images reveal little information. Also, why did Flot2 and Wnt3 travel together intracellularly? Figure 4, supp. Figure 2A shows that wnt3 binds Flot2-GFP expressing cell protrusions. Did the authors describe and discuss that in the text?

5) "We have therefore improved the resolution of the images and added an analysis of AGS cells expressing Ror2-BFP, Flot2-GFP, and membrane-mCherry, which shows the membrane localisation of Flot2 and Ror2 (Supp. Figure 3a)."

This figure cannot be located.

6) "The nuclei and cell boundaries are not clear; the markers for these should be included to give confidence where and how the quantification was conducted. " We have marked the nuclei as well as the adjacent cytoplasm with asterisks – to show the localisation in which the fluorescence of KTR-mCherry was measured.

Labeling did not solve the issue as it is impossible to see some cells' nuclei and cell boundaries.

7) Figure 5:

a) Figure 5A: Flot2 expression doesn't seem to be at the membrane. The staining is very different from Figure 5B, and Figure 5-supplement 1A. Could the authors explain it?

b) Page 8, line 237. "Flot2 resulted in a substantial reduction in Ror2 membrane localisation and its intracellular accumulation". Did the authors want to describe that intracellular accumulation is increased? Additionally, there is significant labeling in intracellular areas in Figure 5A.

c) Supp. Figure 3 cannot be located. The text described in Figure 5E; Supp. Figure 5—figure supplement1B-DE).

8) Figure 6:

a) Figure 6C: The cell is round. Could the authors explain it? How was Wnt8a-mchery expressed?

b) Figure 6J: DNA injection results in mosaic labeling, which is good for imaging, but this is not a reliable method for comparing the functional consequences from such injection.

c) Fig6 K, L: Please specify which groups were used to compare DNFlot2/wnt8a (p<0.001 in K, ns in L). These data were not described in the text.

*Reviewer #3 (Recommendations for the authors):*

The manuscript has improved significantly after the revision and the authors have provided a satisfactory response to the comments. However, the revisions done on a comment were not visible in the revised document.

(Comment Nr. 8) Line 337 -339: Authors claim that they have reworded the statement related to diffusion; "experiments in *Drosophila* suggest that Wg is similarly not freely diffusible as flies with a membrane-tethered form of Wg develop largely normal", however, it remains unchanged in the revised version.

I don't think Alexandre et al. 2014 suggested that endogenous Wg is not freely diffusible. They used an artificially designed NRT-Wg fusion protein to show the maintenance of signaling in the absence of a long-range gradient and a near normal patterning of the wing with delayed development.

---

## [Author Response]

Essential revisions:Reviewer #2 (Recommendations for the authors):Figure 1A: Only single cells are shown to represent each of the cell types assayed. Zoomed-out images are needed for the reader to gauge what extensions look like across the cell population. It is difficult to determine which extensions are considered cytonemes and how they differ from other observable filopodia.

We are grateful for this comment, as it was also raised by reviewer 1. We have, therefore, replaced the images in figure 1 with zoomed-out images to show multiple cells

Figure 1H-I: There are a lot of red dots that do not appear to be associated with cells or filopodia on cells. Is this a background or signal specificity issue?

We have improved and replaced the staining for Myosin-X and Evi with images containing less background. In addition, zoomed-in boxes are overexposed to highlight localisation on cytonemes.

Figure 2B: It is difficult to determine the relative positivity of Wnt signaling activity in the cells assayed because only signal-positive cells are visible. A DAPI co-stain or DIC image is needed to see how many cells are in the field. Same issue for Figure 4E.

HFE-145 cells have basal Wnt activity, so all transfected cells display fluorescence and, as long as transfection efficiencies are consistent, offer a reliable method of counting cells for this reason. Additionally, since this is a co-cultivation, a DAPI stain or DIC image would also stain AGS cells. Therefore, we cannot distinguish between untransfected HFE or AGS cells and therefore reliably include these in the cell count. Whilst this is a limited method, counting cells based on TCF reporter expression is the most reliable for co-cultivation. Furthermore, the BrdU assay and the new colony formation assays were also used to validate further the trends observed using this method, so we are confident in these results.

Figure 2C: How is the relative reporter activity determined? Relative to what? Please provide this information in the results or figure legend.

Reporter activity was calculated by measuring the fluorescence of nuclei and subtracting the background. The average nucleus fluorescence for HFE-145 cells was calculated and set as "1". All other groups were made relative to the HFE-145 control, expressing no constructs other than the reporter. The figure legend for Figure 2 states: "Fluorescence of STF mCherry reporter was measured after 48 hrs and compared to untransfected control".

Lines 99-100: The authors conclude the majority of Wnt3 protein is disseminated by IRSp53 dependent cytonemes in gastric cells. It is not clear this can be concluded from the data shown. Is there a way to specifically monitor cytoneme dependent vs. secretion-based Wnt reporter activation? Does the IRSp534K mutant have any effect on Wnt secretion into culture media?

We do not exclude the mechanism of secretion of Wnts on extracellular vesicles. However, we demonstrate that cytonemes play a substantial role in disseminating Wnt3 protein between gastric cancer cells. To support this finding, we have used a colony-forming assay to show further the positive influence of overexpression of Wnt3 on the ability to form colonies of neighbouring cells can be abrogated by blockage of cytonemes. Furthermore, the synergistic effect of Wnt3 and Flot2 on colony formation can be cancelled by blockage of cytoneme formation (Figure 2, Figure 4, Suppl. Figure 4).

The IRSp534K cells shown in Sup Figure 1A look very unhealthy, so might IRSp534K be having off-target effects – like cell death? The membrane blebs are concerning. Are cells similarly affected upon Cdc42 or N-Wasp inhibition or dominant-negative protein expression?

We agree with the comment of the reviewer. Therefore, we have replaced the previous image with IRSp53^4K^ expressing cells, which are more representative.

Figure 2D: How were HFE-145 cells specifically quantified? Are they labeled differently than the cocultured AGS cells? Please provide a brief explanation how sending vs receiving cells were identified and accounted for in the quantification. Same question for Figure 2E-F.

HFE-145 cells were exclusively expressing the STF-mCh reporter, whilst AGS cells were only transfected with GFP-tagged constructs, therefore, are distinguished by fluorescence. Additional information has been added to the Results section.

Line 117: Is cytonemal a word?

We have changed the word "cytonemal" to "cytoneme-mediated" or just "cytoneme", depending on the context.

Figure 3B: The extensions in HFE-145 that are increased upon Flot2 expression look more cytoneme-like than the extensions in AGS cells. Many of the AGS cell extensions look like retraction fibers. How are cytonemes being distinguished from retraction fibers? Please indicate in the figure what is a cytoneme and what is not. Were only cytoneme-like extensions quantified in C-E?

We have replaced images in Figure 3B with zoomed-out ones to include more cells and to show a broader representation of the protrusions. The protrusions measured in this figure are referred to as "filopodia" due to our inability to distinguish filopodia from cytonemes from only membrane markers. Morphological easy identifiable extraction fibres were not counted in the quantification, but all filopodia were counted.

Figure 4B: Line 143 states that the authors want to determine whether Flot2 is specific to Wnt cytonemes and not just filopodia, so they look for Wnt3 and Flot2 localizing to the same extensions. Line 147 states that Wnt3 and Flot2 colocalize on cytonemes, but the wording is vague. Is the suggestion that Wnt3 and Flot2 are colocalizing as a protein complex, or is it that they are both present in the same cytonemes but not necessarily in complex? If the former, please provide colocalization coefficients. Same question for 4D. Even if Wnt3 and Flot2 do colocalize, this will not confirm Flot2 is dedicated to Wnt cytonemes. To make this conclusion, the authors will need to determine whether Flot2 does or does not colocalize with other signaling proteins that are transported along cytonemes. The authors cannot rule out that Flot2 is a general modulator of cytoneme transport.

We suggest from the data in Figure 4 that Flot2 and Wnt3 co-localise on cytonemes. However, this does not necessarily mean they are in complex together, and this co-localisation is investigated in the context of Ror2 later in the paper. Therefore, we have added a statement to clarify: "Whether Flot2 and Wnt3 physically interact as part of a complex requires further clarification".

Figure 5A: Quantification of colocalization is needed. Figure 5B: Does the arrow indicate a Flot2 punctum in a cytoneme, or is it simply indicating the cytoneme?

We have calculated the PCC and added this value to the Figure legend 5. Arrow Flot2/Ror2 co-localisation on cytonemes. This information has been added to the Figure legend 5.

Figures 5C-D: Quantification of changes in cytoneme formation and images of more than one cell for each condition are needed to gauge the extent to which cellular extensions are impacted by Ror2 and Flot2 gain/loss.

We agree with the reviewer and have changed Figure 5E, F and focused on the localisation of Ror2 in the Golgi. Suppl. Figure 3 shows more representative images from examples with the other organelle markers.

Figure 5I: Please add arrows to indicate examples of what were counted as cytonemes vs. non-cytoneme cell extensions that are present.

Cytonemes are defined as cellular protrusion transporting signalling components (Kornberg et a., 2014; Zhang and Scholpp, 2019). Therefore, cytonemes and the presence of Wnt3 can be defined by the presence of other Wnt signalling components, such as Ror2 (Mattes et al., 2018). Therefore, we call these protrusions in Fig5i cytonemes due to the presence of Ror2

Reviewer #3 (Recommendations for the authors):1) Line 112: authors say "that co-expression of Wnt3 together with IRSp534k led to a significant downregulation of actively proliferating cells" however, quantification in fig2F appears to show a very weak effect and whether the difference between +Wnt3 and +Wnt3/IRSp534K is significant is not clear. The p-values comparing various treatments (as given in 2d) should also be given for 2f.

We have added comparative p values between groups to Figure 2F. We have also corrected the text, as the difference between Wnt3 and Wnt3/IRSp53-4K is not significant but close to significance (p=0.08), and we speculate the reasoning behind this.

2) Line 143: The authors say that they investigated if Flot2 specifically regulates Wnt3 cytonemes. However, they only show that Flot2 and Wnt3 colocalize on cytonemes, which does not directly suggest that Flot2 perturbation will only affect Wnt3 containing cytonemes.

We agree with this comment and have changed the word "specifically" to "actively" to clarify that we are not claiming Flot2 does not regulate other Wnt proteins or cytonemes, but that it is involved in Wnt3 cytonemes rather than generally affecting filopodia.

3) Figure 4: It is intriguing to see that treatment of the control AGS cells with ΔN-Flot2, did not completely abolish the Wnt signalling activity or cell numbers/BrDU incorporation; even though it led to a significant reduction in the cytoneme numbers and length (Figure 3). As the GC cells are supposed to have higher Wnt3, which is suggested to enhance their proliferation, is it possible that the lack of a strong phenotype could be because of alternative mechanisms of Wnt3 release which remained unaffected by ΔN-Flot2 treatment?

We agree with the reviewer that – in addition to cytonemes – other mechanisms such as Wnt3 on exosomes could be important for ligand dissemination. Therefore, we have added the following sentence to reflect this point raised: "We cannot rule out that the modest inhibition of paracrine Wnt signal activation and proliferation by ∆N-Flot2-GFP at endogenous Wnt3 levels may be due to secretion of Wnt3 via cytoneme-independent mechanisms. Alternatively, compensation by Flot1 may still transport low levels of Wnt3 but not higher levels seen with Wnt3 overexpression."

4) Figure 5C and d are quite confusing. The text in lines 189-190 says "Consistently, expression of a dominant-negative form of Flot2 resulted in a strong reduction of cytonemes (Figure 5c) " while Figure 5C is marked as Flot2 siRNA. Mentioned in line 180, "we observed a similar phenotype after knock-down of Flot2 – a reduction of cytonemes and removal of Ror2 from the membrane (Figure 5d)" while Fig5d is about DN-Flot2.

We have corrected the text's figure references and reworded them to make them clearer.

5) Line 283: I think authors should rephrase the statement about *Drosophila* Wg being not diffusible, as there are several studies that have shown the long-range movement of endogenous Wg via a collection of extracellular carriers. Whereas the function of membrane-tethered Wg by JP Vincent group was shown via an artificially made Wg fusion protein and not for endogenous Wg.

We have reworded the statement about diffusion.

6) Line 339-341: With the statement "Consistently, we found that ectopic Flot2 expression leads to an effective transport from Ror2 to the plasma membrane and thus, to enhanced Wnt3 cytoneme formation in the source cells (Figure 5)" authors are implying that they found Ror2 to be the mediator of Wnt3 transport on cytonemes. However, the experiments analysing the direct effect of Ror2 manipulation on Wnt3 transport have not been shown in Figure 5 and this conclusion appears to be based on their interaction shown by other studies.

We have removed the word "Wnt3" from this sentence as we have not explicitly evaluated the percentage of these cytonemes which are Wnt3-positive or bearing other Wnt proteins, which is likely since AGS cells express multiple Wnt proteins.

Other comments:Fig3E: Y-axis label should be added.

A label for the Y-axis on Figure 3E has been added.

Line170: "whereas the Wnt3-induced signalling was attenuated by co-expression with ΔN-Flot2-GFP". This sentence is confusing as the data in Figure 4g is about the cell number and not signalling readout.

We have replaced the word "signalling" with "proliferation."

Line 171: "Similarly, IRSp534L attenuated increases seen in the presence…" appears to be an incomplete sentence.

We have corrected this, and the sentence now reads, "Similarly, IRSp53-4K attenuated cell number increases…"

[Editors' note: further revisions were suggested prior to acceptance, as described below.]

The manuscript has been greatly improved but there are a few remaining issues that need to be addressed:Reviewer #1 (Recommendations for the authors):There are a few issues remain that should be addressed.General issues:1. The naming system of the supplementary figures is very confusing. Most supplementary figures are not labeled, and some figures were mislabeled. The J editorial support team suggested downloading each figure and renaming them according to the file title. Even by doing that, many figures cannot be located. It seems that supplementary figures are associated with the main figures. The supp Figure 1 for Figure 2 is named Figure 2 supp Figure 1. Two supplementary figures for Figure 4, designated as Figure 4 suppl Figure 1, and Figure 4 suppl Figure 2. These names were used in the main text but not in the response letter.For example,1) Supp. Figure 1E: seems to be Figure 3 suppl figure 1C.2) Supp. Figure 1C: seems to be Figure Suppl Figure 1A3) Supp. Figure 2D: I found this panel in Figure 3 suppl figure 1D.4) Supp. Figure 2E: cannot be located.5) Figure 6D, E; Supp. Figure 3d: Supp. Figure 3d was not found.6) One supplemental figure is labeled as supplementary figure 6. Is it correct?Please review the manuscript, label all figures, and cite figures properly. Otherwise, the manuscript is impossible to be understood.

We apologize for the confusing labelling in the ‘Responses to reviewers‘ -file. We have updated the labelling for the figure supplements as suggested. For example, we ordered them and cited them sequentially, e.g., Figure 1, Figure 2, Figure 2—figure supplement 1, Figure 2—figure supplement 2, Figure 3, Figure 3—figure supplement 1. We have now changed all names accordingly in all documents. We have also added the names to the PDFs for clarification.

2. The responses were extensive, but the authors did not describe or label where the changes were, which makes the evaluation very challenging.

We have marked all changes in red in the accompanying manuscript file ‘Manuscript with changes’.

Specific issues:The following issues were not addressed adequately:1) The authors did not address this issue: "Figure 2D shows Relative number of HFE cells per image".Did the authors use the number from the control as the base? Is it more meaningful if the actual values are used?

We report relative cell numbers to highlight what percentage of cell number (whether higher or lower) changed from the control cells. As a basis, we used the cell number of HFEs as shown as the first data set in Figure 2D (ctrl).

Additionally, is it possible to image Wnt3 is being transported to the receiving cells?"The authors did not address if a video is provided.

We have added a video (Figure1-Video1) showing the transport of Wnt3 in cytonemes.

The Wnt3 staining in Figure 2 Suppl Figure 1A is messy. Are all GFP staining Wnt3 signals? Where was Wnt3 expressed? Which cells are the recipient cells?

In Figure2—figure supplement 1A, we show an antibody staining against endogenous Wnt3 in a co-cultivation of STF-mCh-expressing HFE-145 cells with AGS cells (ctrl). The recipient cells express the STF-mCherry reporter. HFE-145 cells increase the expression of the STF-mCh reporter after co-cultivation with Wnt3-positive AGS (Figure 2A, B).

2) "Figure 2E: The BrdU stains are brightfield images and do not use any fluorescence. Therefore, there are no channels to split. "Could authors show large images for better visualization if this is the case?

We have uploaded high-resolution images, and it is up to the reviewer to zoom in and appreciate the staining details.

3) "we performed antibody staining against Wnt3 after Flot2 KD (and in DN-Flot2-expressing cells) to assess its localisation (Supp. Figure 2)."This figure cannot be located. Is it Figure 3, suppl Figure 1C?

This mistake has been corrected (see comments above). The correct labelling is Figure4—figure supplement 1C

4) "For clarification, we have added time-lapse images of Flot2-GFP/Wnt3-mCh expressing cells (Supp. Figure 2G), where Flot2 and Wnt3 can be seen travelling together intracellularly."Is it Figure 4, supp. Figure 2B? This figure cannot be located. Time-lapse was not provided. The still images reveal little information. Also, why did Flot2 and Wnt3 travel together intracellularly? Figure 4, supp. Figure 2A shows that wnt3 binds Flot2-GFP expressing cell protrusions. Did the authors describe and discuss that in the text?

This mistake has been corrected (see comments above). The correct labelling is Figure4—figure supplement 1B. Our data suggest that Wnt3 and Flot2 co-localise at the plasma membrane (Figure 4B-D), and we also find co-localisation intracellularly (Figure4—figure supplement 1A). Furthermore, wnt3-mCherry and Flot2-GFP can be observed together in moving clusters within a cell (Figure4—figure supplement 1A, B). This has been explained in the main text (Line 174-182).

5) "We have therefore improved the resolution of the images and added an analysis of AGS cells expressing Ror2-BFP, Flot2-GFP, and membrane-mCherry, which shows the membrane localisation of Flot2 and Ror2 (Supp. Figure 3a)."This figure cannot be located.

This mistake has been corrected (see comments above). The correct labelling is Figure5—figure supplement 1A.

6) "The nuclei and cell boundaries are not clear; the markers for these should be included to give confidence where and how the quantification was conducted. " We have marked the nuclei as well as the adjacent cytoplasm with asterisks – to show the localisation in which the fluorescence of KTR-mCherry was measured.Labeling did not solve the issue as it is impossible to see some cells' nuclei and cell boundaries.

We have encircled the nuclei and the corresponding cytoplasm for easier visualization, as suggested by the reviewer in Figure 5. Cells which overlap with other cells or cells in which the cytoplasm and nucleus could not be identified have been excluded.

7) Figure 5:a) Figure 5A: Flot2 expression doesn't seem to be at the membrane.

We respectfully disagree with the reviewer. Clusters of endogenous Flot2 and Ror2 can be found at the plasma membrane (Figure 5A, arrows).

The staining is very different from Figure 5B, and Figure 5-supplement 1A. Could the authors explain it?

In Figure 5A, we show the localisation of the endogenous proteins, whereas B shows the localisation of the overexpressed constructs of Ror2-BFP and Flot2-GFP. The labelling of the figures in B, C, and D has been changed to highlight the differences.

b) Page 8, line 237. "Flot2 resulted in a substantial reduction in Ror2 membrane localisation and its intracellular accumulation". Did the authors want to describe that intracellular accumulation is increased? Additionally, there is significant labeling in intracellular areas in Figure 5A.

Indeed, the siRNA-mediated knock-down of Flot2 leads to a relocation of Ror2 from the plasma membrane (Figure 5B) to intracellular sites (Figure 5C, arrows). This observation is supported by the data shown in Figure 5D, in which Ror2 is similarly localised to intracellular areas after co-expression of a dominant-negative form of Flot2. The reviewer is correct that in Figure 5A, Ror2 and Flot2 can be found at the membrane and intracellular sites.

c) Supp. Figure 3 cannot be located. The text described in Figure 5E; Supp. Figure 5—figure supplement1B-DE).

This mistake has been corrected (see comments above). The correct labelling is Figure5—figure supplement 1B-E.

8) Figure 6:a) Figure 6C: The cell is round. Could the authors explain it? How was Wnt8a-mchery expressed?

The early gastrula cells are mesenchymal cells that can change their form quickly between a flat morphology and a round morphology. Wnt8a-mCherry was expressed as a DNA construct described in the Methods and Materials section, ‘Zebrafish microinjections and image analysis’.

b) Figure 6J: DNA injection results in mosaic labeling, which is good for imaging, but this is not a reliable method for comparing the functional consequences from such injection.

We respectfully disagree with the reviewer. Mosaic labelling is an essential aspect of this approach. By doing so, we generate small, individual cell clones which can act as focal Wnt8a sources. We alter the spreading of Wnt8a from these small sources by changing the emergence of cytonemes. Reduced cytoneme formation partially rescues the AP phenotype after focal activation of Wnt8a.

c) Fig6 K, L: Please specify which groups were used to compare DNFlot2/wnt8a (p<0.001 in K, ns in L). These data were not described in the text.

In Fig6K, the ctrl group has been used as the basis for the T-Test. In addition, we have compared Flot2/wnt8a with dN-Flot2/wnt8a. Finally, we have added the test results to the figure.

As suggested by the reviewer, we have performed a Pearson’s χ2 test in Fig6L. As stated in the figure legend, the test revealed a significant difference between the ctrl group (expected) and experimental groups (observed) with 2 degrees of freedom (df) and a p-value < 0.05 of χ2 <0.001. In addition, we compared *flot2* injected embryos to the other groups. This test revealed a significant difference between the *wnt8a* group and the *flot2+wnt8a* group, but not to *wnt8+ΔN-flot2*. We have clarified these test results in the figure.

Reviewer #3 (Recommendations for the authors):The manuscript has improved significantly after the revision and the authors have provided a satisfactory response to the comments. However, the revisions done on a comment were not visible in the revised document.(Comment Nr. 8) Line 337 -339: Authors claim that they have reworded the statement related to diffusion; "experiments in *Drosophila* suggest that Wg is similarly not freely diffusible as flies with a membrane-tethered form of Wg develop largely normal", however, it remains unchanged in the revised version.I don't think Alexandre et al. 2014 suggested that endogenous Wg is not freely diffusible. They used an artificially designed NRT-Wg fusion protein to show the maintenance of signaling in the absence of a long-range gradient and a near normal patterning of the wing with delayed development.

We have changed the text: ‘In support of these observations, experiments in *Drosophila* using an artificially membrane-tethered form of Wg showed that free diffusion is not necessary, as without the ability of free diffusion, *Drosophila* wing patterning was almost completely unaffected (Alexandre, Baena-Lopez and Vincent, 2014).